# KIS counteracts PTBP2 and regulates alternative exon usage in neurons

Marcos Moreno-Aguilera[1†], Alba M Neher[1†], Mónica B Mendoza[1], Martin Dodel[2], Faraz K Mardakheh[2], Raúl Ortiz[1], Carme Gallego[1]*

[1]Molecular Biology Institute of Barcelona (IBMB), CSIC, Barcelona, Spain; [2]Barts Cancer Institute, Queen Mary University of London, London, United Kingdom

**Abstract** Alternative RNA splicing is an essential and dynamic process in neuronal differentiation and synapse maturation, and dysregulation of this process has been associated with neurodegenerative diseases. Recent studies have revealed the importance of RNA-binding proteins in the regulation of neuronal splicing programs. However, the molecular mechanisms involved in the control of these splicing regulators are still unclear. Here, we show that KIS, a kinase upregulated in the developmental brain, imposes a genome-wide alteration in exon usage during neuronal differentiation in mice. KIS contains a protein-recognition domain common to spliceosomal components and phosphorylates PTBP2, counteracting the role of this splicing factor in exon exclusion. At the molecular level, phosphorylation of unstructured domains within PTBP2 causes its dissociation from two co-regulators, Matrin3 and hnRNPM, and hinders the RNA-binding capability of the complex. Furthermore, KIS and PTBP2 display strong and opposing functional interactions in synaptic spine emergence and maturation. Taken together, our data uncover a post-translational control of splicing regulators that link transcriptional and alternative exon usage programs in neuronal development.

*For correspondence:
cggbmc@ibmb.csic.es

†These authors contributed equally to this work

Competing interest: The authors declare that no competing interests exist.

## Editor's evaluation

This well-presented and concise study provides important new insights into the regulation of alternative splicing in neurons by the kinase KIS. The provided data convincingly demonstrates how KIS-mediated phosphorylation of the splicing regulator PTB2 controls exon usage during neuronal differentiation. The findings will be of interest to investigators of splicing regulation, neuronal differentiation and maturation.

## Introduction

In the mammalian brain, alternative splicing (AS) is a key mechanism that contributes substantially to the enormous molecular diversity among neuronal cell types (*Barbosa-Morais et al., 2012*; *Mazin et al., 2021*; *Merkin et al., 2012*). Despite the similarity of neuronal gene expression programs, each neuronal subclass can be distinguished by unique alternative mRNA processing events. Moreover, AS is known to drive many aspects of neurogenesis, including neural progenitor cell proliferation and differentiation, axon guidance, synapse formation, and synaptic plasticity (*Vuong et al., 2016b*; *Furlanis and Scheiffele, 2018*). The relevance of AS as a regulatory mechanism is evidenced by numerous examples in which alterations in splicing are associated with pathological states (*Gandal et al., 2018*; *LaForce et al., 2023*; *Schieweck et al., 2021*). Although fine-tuning mRNA processing is essential for neuronal activity and maintenance, insights into its regulation are just starting to emerge. RNA-binding proteins (RBPs) play a critical role in regulating AS patterns. These proteins bind to pre-mRNAs and form ribonucleoprotein (RNP) complexes that can affect splicing by influencing the recognition of splicing signals or by blocking access of the splicing machinery to certain regions of

the pre-mRNA. Some RBPs act as splicing enhancers, promoting the inclusion of specific exons, while others act as splicing repressors inhibiting the inclusion of exon inclusion (*Fisher and Feng, 2022*; *Grosso et al., 2008*; *Traunmüller et al., 2023*).

Among the most investigated RBPs are the polypyrimidine tract-binding proteins, PTBP1 and PTBP2, which are considered master regulators of neuronal fate (*Keppetipola et al., 2012*; *Linares et al., 2015*; *Zhang et al., 2019*). These two proteins exhibit tissue-specific expression patterns in which PTBP1 is expressed in most cell types and neuronal progenitor cells, whereas PTBP2 is expressed primarily in differentiating neurons and testis. During the differentiation of neuronal progenitor cells to postmitotic neurons a switch takes place from the predominant expression of PTBP1 to its neuronal paralog PTBP2, which is essential for the stem cell to neuron transition. PTBP2 null mice die shortly after birth and exhibit misregulation of AS in genes involved in cytoskeletal remodeling and cell proliferation (*Boutz et al., 2007*; *Licatalosi et al., 2012*) as well as in neurite growth and synaptic transmission (*Li et al., 2014*). Furthermore, PTBP2 knockout brains display reduced neural progenitor pools and premature neurogenesis (*Licatalosi et al., 2012*). PTBP2 also plays an important role in the embryonic-specific repression of alternative exons. As PTBP2 expression decreases from its highest levels in the embryonic brain to moderate levels during postnatal development, a cohort of exons switches from being skipped in embryos to displaying enhanced inclusion in adults (*Licatalosi et al., 2012*; *Li et al., 2014*; *Zheng et al., 2012*). Thus, sequential downregulation of PTBP1 followed by PTBP2 contributes to the activation of neural exon networks at the appropriate stages of development (*Raj and Blencowe, 2015*). AS of the *DLG4* transcript, which encodes the excitatory postsynaptic protein PSD95, is an example of the relevance of the PTBP1/2 switch in neuronal differentiation (*Zheng, 2016*). Skipping of exon 18 in *DLG4* transcripts causes premature translation termination and nonsense-mediated mRNA decay (NMD). The controlled induction of this exon during neuronal development is essential to ensure the timing of synapse formation (*Zheng et al., 2012*). This exon may also be dynamically regulated in mature neuronal circuits to adapt PSD95 expression to the requirements of synaptic remodeling (*Lopez Soto et al., 2019*).

PTBP1 and PTBP2 are 74% identical in amino acid sequence and have a similar domain organization of four RNA recognition motifs (RRMs) connected by three linker regions. Although primarily characterized as repressive splicing regulators, they can also activate some splice sites as a function of binding position relative to regulated exons (*Corrionero and Valcárcel, 2009*; *Llorian et al., 2010*; *Xue et al., 2009*). It has been shown that PTBP1/2 RRM2, along with the following linker sequence, is sufficient for splicing repressor activity (*Robinson and Smith, 2006*). RRM2 can interact with both RNA via its canonical β-sheet surface, and with other co-regulators such as Matrin3, Raver1, and hnRNPM (*Coelho et al., 2015*). Furthermore, Tyr247 of PTBP1 (PTBP2 Tyr244) within RRM2 is particularly critical for co-regulatory protein interactions (*Joshi et al., 2011*; *Rideau et al., 2006*). Regarding the regulation of PTBP proteins, different phosphate modifications are located in the unstructured regions, including the N-terminal, Linker 1, and Linker 2 regions (*Pina et al., 2018*). To date, the kinases involved are still under investigation as well as the role of these post-translational modifications in controlling PTBPs splicing activity.

KIS (encoded by the gene *UHMK1*) was first identified as a Kinase that Interacts with Stathmin (*Maucuer et al., 1997*), a phosphoprotein that controls microtubule dynamics (*Watabe-Uchida et al., 2006*). Curiously, KIS is the only known protein kinase that possesses a U2AF homology motif (UHM), an atypical RRM that has lost its RNA-binding ability, but mediates protein–protein interactions at the C-terminal region of KIS (*Corsini et al., 2007*; *Kielkopf et al., 2004*; *Manceau et al., 2008*). This motif shares 42% sequence similarity with U2AF65, a 65-kDa subunit of the splicing factor U2AF, suggesting that KIS could be involved in splicing-related processes. Various lines of evidence support the function of KIS in the modulation of gene expression at different levels such as transcription, splicing, mRNA stability, and protein translation (*Arfelli et al., 2023*; *Manceau et al., 2008*; *Pedraza et al., 2014*). Although KIS is ubiquitously expressed, higher levels are detected in the nervous system, and KIS protein is gradually upregulated during postnatal development, reaching its highest level in the mature brain (*Bièche et al., 2003*). In mature neurons, KIS has been implicated in dendritic spine morphogenesis and synaptic activity (*Pedraza et al., 2014*).

Here, we find that KIS kinase is involved in the control of alternative exon usage, a key mechanism for promoting functional gene expression patterns during neuronal differentiation. We show that AS regulators are the main phosphorylation targets of this UHM-containing kinase. Notably, we find

that PTBP2, an essential AS factor for neuronal maturation, is the major phosphotarget of KIS. We demonstrate that phosphorylation by KIS dissociates PTBP2 protein complexes affecting their ability to bind RNA and therefore inhibiting PTBP2 activity. Furthermore, KIS counteracts PTBP2-mediated inhibition of dendritic spine maturation in a phosphosite-dependent manner, and downregulation of KIS partly rescues the negative effects caused by decreasing PTBP2 levels on spine maturation. Taken together, our data point to the notion that KIS phosphorylation counteracts PTBP2 activity and, in this way, alters the expression patterns of gene isoforms involved in neuronal differentiation and synaptic activity.

## Results

### KIS kinase regulates exon usage during neuronal differentiation

KIS contains a UHM domain frequently found in splicing regulatory proteins, and a GFP–KIS fusion accumulates in nuclear substructures adjacent to those formed by splicing factors (*Figure 1A*). Prompted by its role in proper expression of key postsynaptic proteins (*Pedraza et al., 2014*), we decided to analyze genome-wide exon usage during neuronal differentiation in vitro. First, immature neurons from mouse primary cortical neurons of 7 days in vitro (DIV) were compared with mature neurons from 18 DIV cultures when expression of *UHMK1*, the gene coding for KIS, reached maximal levels in cortical cultures (*Figure 1B*). Second, cortical neurons were infected at 7 DIV with a lentiviral construct co-expressing green fluorescent protein (GFP) and an shRNA targeted against *UHMK1* (shKIS) and collected at 18 DIV (*Figure 1C*). Total RNA was purified from triplicate samples for deep sequencing analysis. When compared to control (shCtrl), shKIS-infected cells displayed a fivefold reduction in *UHMK1* mRNA levels, similar to that at onset of neuronal differentiation (*Figure 1D*). Our analysis revealed that the impact of KIS on exon usage affected both exon inclusion and skipping, although with a clear bias toward exon inclusion 6.3% vs 2% skipping (*Figure 1E*). Notably, genes containing more than one exon with at least a twofold change coded for proteins significantly enriched in the synapse (*Figure 1—figure supplement 1A*), including two important activators of neurite outgrowth, CAMSAP1, and NCAM1 (*Figure 1—figure supplement 1B*). These data suggest that KIS promotes exon inclusion in genes related to neuronal differentiation, especially genes encoding synaptic proteins. Supporting this notion, the top 100 exons upregulated by KIS, that is downregulated by KIS knockdown, were strongly enriched among those increasingly used by cortical neurons from 7 to 18 DIV (*Figure 1F*). Among a total of 8457 exons that displayed low variability (Fano factor <5 × 10⁻⁴) within the KIS knockdown triplicate samples, 1141 and 1106 exons were found significantly upregulated and downregulated, respectively, by KIS (*Figure 1G*; *Supplementary file 1*). In addition, 28.0% of exons upregulated by KIS were also upregulated during differentiation, whereas only 18.8% decreased their usage from 7 to 18 DIV (*Figure 1—figure supplement 1C*). The opposite behavior was observed for exons downregulated by KIS in which 14.3% and 24.8% increased or decreased their usage during differentiation, respectively (*Figure 1—figure supplement 1C*). Finally, whereas genes containing KIS-downregulated exons did not retrieve specific gene ontology (GO) terms, exons upregulated by KIS encoded a higher percentage of protein disorder (*Figure 1—figure supplement 1D*) and were found in genes preferentially coding for RBPs (*Figure 1H*), key factors of RNA granule formation and transport along neurites. Overall, our data link KIS to the control of exon usage in cortical neurons and suggest a pivotal role of this UHM-containing kinase as a regulator of AS during neuronal differentiation.

### Phosphoproteomics links KIS to RNA splicing and identifies PTBP2 as a novel phosphotarget

One possible way to control the use of alternative exons during neuronal differentiation is by changing the expression of splicing regulators. Our RNA-seq analysis showed that KIS knockdown did not affect transcript levels of splicing regulators whose expression underwent changes during cortical neurons differentiation (*Figure 2—figure supplement 1A*). Therefore, we tested whether the KIS phosphoproteome could reveal the mechanism by which KIS regulates exon usage. For this, we analyzed phosphorylation changes in the proteome of HEK 293T cells expressing KIS and KIS$^{K54A}$, a point mutant without kinase activity (*Maucuer et al., 1997*). We chose this cell line because of its high transfection efficiency and, although it is an embryonic kidney-derived line, there is evidence to suggest a neuronal

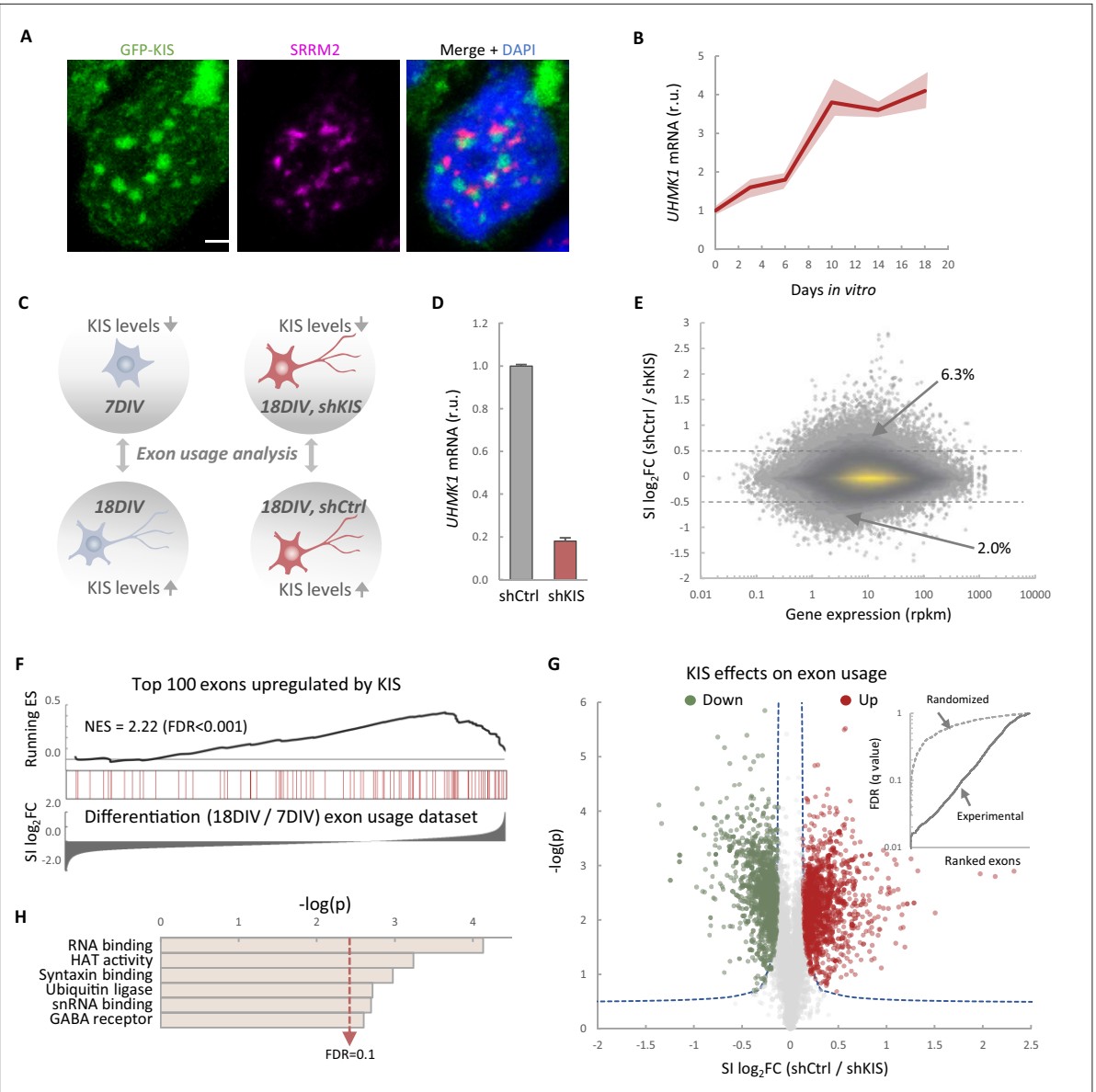

**Figure 1.** KIS promotes the inclusion of exons upregulated during neuronal differentiation. (**A**) Immunofluorescence staining of endogenous speckle marker SRRM2 (magenta) and 4',6-diamidino-2-phenylindole (DAPI) (bue) in HEK293T cells transfected with GFP-KIS (green). Scale bar, 2 µm. (**B**) Differential expression of *UHMK1*, the gene encoding KIS, in cultured cortical neurons from E17.5 mouse embryos. Samples were collected at different days in vitro (DIV), analyzed by qRT-PCR and made relative to day 0. Mean ± standard error of the mean (SEM) (*n* = 3) are plotted. (**C**) Experimental design of exon usage analysis. Cultured cortical neurons from E17.5 mice were collected at 7 or 18 DIV (left experimental set). Cortical neurons were transduced at 7 DIV with lentiviral vectors expressing shCtrl or shKIS and collected at 18 DIV (right experimental set). (**D**) *UHMK1* mRNA levels in primary cortical neurons transduced at 7 DIV with lentiviral vectors expressing shCtrl or shKIS and collected at 18 DIV. Bars represent mean ± SEM (*n* = 3) (**E**) MA-plot shows splicing index (SI) fold change (log$_2$FC) as a function of gene expression in rpkm (reads per kilobase million) in control neurons (shCtrl) compared to KIS knockdown neurons (shKIS) as in panel D. The percentage of exons displaying a log$_2$FC above 0.5 or below −0.5 is indicated. (**F**) RNA-seq analysis from cortical neurons expressing shCtrl or shKIS as in panel D and from cortical neurons collected at 7 and 18 DIV for neuronal differentiation (*n* = 3). Barcode plot shows the position of exons upregulated by KIS, that is downregulated by KIS knockdown, within the transcriptomic dataset of exons upregulated during neuronal differentiation. Normalized enrichment scores (NES) and the corresponding false-discovery rate (FDR) values are shown. (**G**) Volcano plot shows *t*-test significance (−log$_2$p) as a function of SI fold change (log$_2$FC) in control neurons (shCtrl) compared to KIS knockdown neurons (shKIS). Significantly (FDR <0.05) downregulated (*n* = 1106, green) or upregulated (*n* = 1141, red) exons are highlighted. Inset shows the dependence of FDR on exon rank from experimental and randomized datasets. (**H**) Gene ontology (GO) term enrichment analysis of genes with exons upregulated by KIS. FDR value is shown.

The online version of this article includes the following figure supplement(s) for figure 1:

**Figure supplement 1.** KIS promotes exon inclusion in genes encoding proteins related to synaptogenesis and neurogenesis.

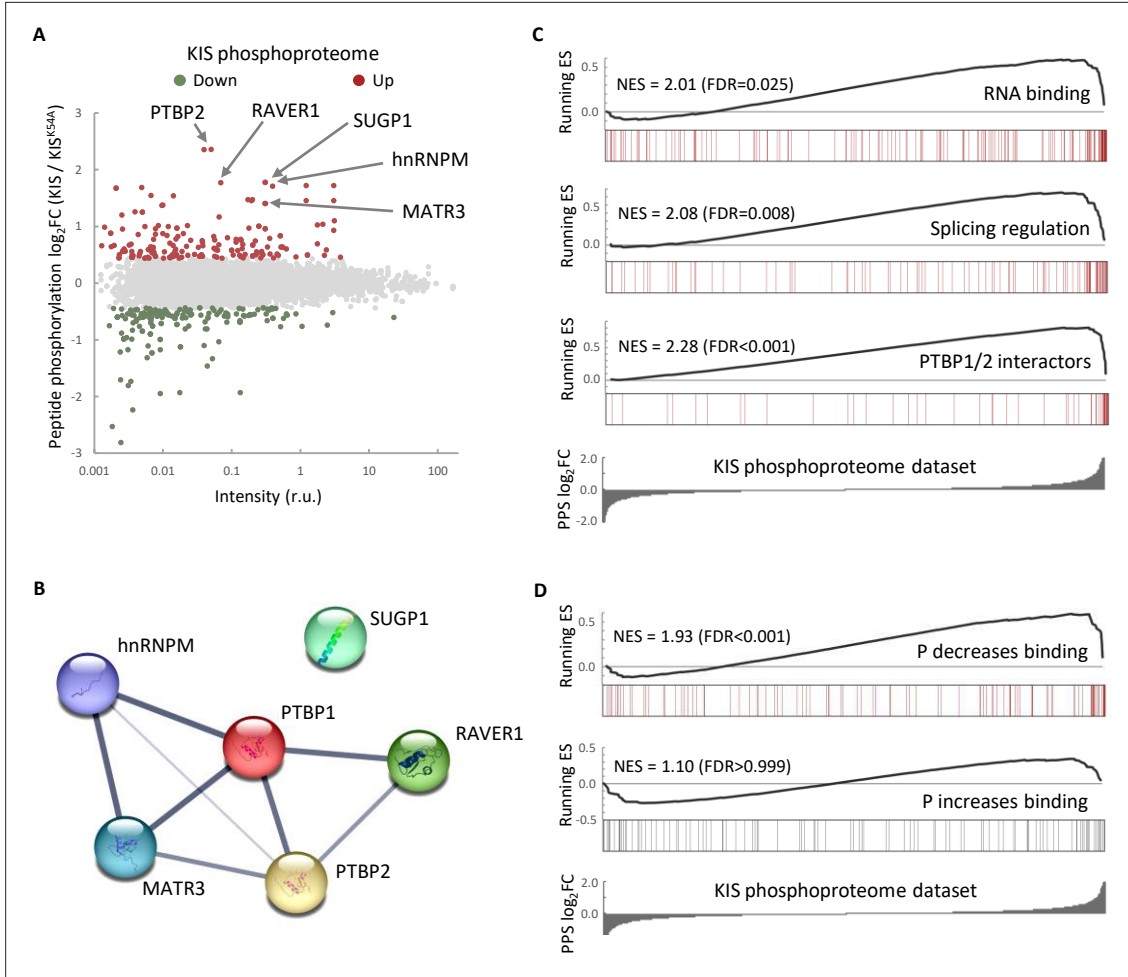

**Figure 2.** The phosphoproteome of KIS identifies splicing regulators. (**A**) MA-plot of phospho-sites found in the proteome of HEK293T cells transfected with KIS and KIS^K54A mutant (see also *Supplementary file 3* for details). (**B**) STRING interaction map of top KIS-phosphorylated proteins. (**C, D**) Gene enrichment analysis of the KIS phosphoproteomic dataset. Barcode plots indicate the rank position of genes from (**C**) Gene ontology (GO) terms and PTBP1/2 interactors and (**D**) phosphorylated RNA-binding proteins (*Vieira-Vieira et al., 2022*) that were found significantly enriched within the KIS phosphoproteomic dataset. Normalized enrichment scores (NES) and the corresponding false-discovery rate (FDR) values are shown.

The online version of this article includes the following source data and figure supplement(s) for figure 2:

**Figure supplement 1.** In vitro phosphorylation of PTBP2 by KIS.

**Figure supplement 1—source data 1.** KIS in vitro kinase assays.

lineage, which could explain the expression of neuron-specific genes (*Lin et al., 2014*). Quantitative phosphoproteomic analysis resulted in the identification of 9994 phospho-sites in a total of 2743 identified proteins (*Supplementary file 2*). As shown in the MA-plot (*Figure 2A*), proteins displaying the strongest differential phosphorylation by KIS are highly related to AS, that is PTBP2, RAVER1, hnRNPM, and MATR3. PTBP2 is considered one of the pivotal splicing regulators in neuronal development (*Keppetipola et al., 2012*) and, strikingly, phosphorylation at serine 308 of PTBP2 is among the top phosphosites in our KIS phosphoproteome (log$_2$FC = 2.36). Furthermore, the kinase assay confirmed that among the three KIS serine targets on PTBP2 (Ser178, Ser308, and Ser434), Ser308 is key for PTBP2 phosphorylation by KIS (*Figure 2—figure supplement 1B–D*). The different levels of reduction in ^32P incorporation displayed by the single phosphonull mutants (*Figure 2—figure supplement 1C*) suggests that phosphorylation follows a hierarchical pattern, P-Ser308 facilitating phosphorylation of the other two phosphosites. Ser178 is present in a very large tryptic peptide (37 aa) that was not detected in the KIS phosphoproteome but, as it fulfills the consensus site phosphorylated by KIS (*Figure 2—figure supplement 1E*), we decided to include this residue in our analysis. Note that the

interactions between splicing factors that are KIS targets have been already shown except for SUGP1 (*Figure 2B* and see also Discussion).

Next, to perform gene set enrichment analysis (GSEA), we ranked KIS targets using a Phospho Protein Score (PPS) that incorporates the fold change of all identified phosphopeptides per protein (see Methods). Proteins that are more phosphorylated in the presence of KIS have higher PPS, while those that are more phosphorylated in the presence of KIS^K54A have lower PPS. GSEA analysis confirmed that KIS-phosphorylated proteins are enriched in regulation of splicing, RNA binding, and PTBP1/2 interactors (*Figure 2C*). Interestingly, when we compared our dataset to a recent RNA-interactomic study, in which the pull-down efficiencies of phosphorylated and non-phosphorylated forms of mRNA-binding proteins were analyzed (*Vieira-Vieira et al., 2022*) (see also Methods), we found that among KIS-phosphorylated proteins there is a significant enrichment of RBPs that decrease their binding to mRNA upon phosphorylation (*Figure 2D*), but not the contrary, suggesting that KIS phosphorylation would likely act by promoting dissociation of protein complexes from mRNA. In all, KIS phosphoproteomics reveals a strong association of this kinase with AS regulators, PTBP2 being a relevant phosphotarget in neuronal differentiation.

## KIS kinase activity counteracts PTBP2 function on exon usage

To define the role of KIS in AS in neurons, we decided to focus our study on the link between PTBP2 and KIS. It is known that loss of PTBP2 causes extensive changes in AS in brain (*Vuong et al., 2016b*). To assess whether these changes may be modified by KIS expression, we performed an enrichment analysis of PTBP2-regulated exons in our cortical samples to obtain population-wide data. We found that PTBP2-inhibited exons are significantly enriched (false-discovery rate [FDR] = 0.001) among KIS-activated exons, supporting the notion that KIS acts on AS, at least in part, by inhibiting PTBP2 activity. This analysis was also carried out with PTBP2-activated exons, but no significant enrichment was observed in this case (*Figure 3A*). As expected, randomization of our original RNA-seq dataset resulted in a total loss of enrichment (*Figure 3—figure supplement 1A*). In addition, applying our exon usage analysis to RNA-seq data from PTBP2 KO cells (*Vuong et al., 2016b*) produced high enrichment scores for both PTBP2-inhibited and activated exons as expected (*Figure 3—figure supplement 1B*), which supports our approach to analyzing exon inclusion from RNA-seq data.

To validate the effect of KIS on PTBP2 activity, we analyzed the AS of *CAMK2B* transcript in KIS knockdown samples (shCtrl vs shKIS). This transcript, which has important neuronal functions, undergoes exclusion of exons 13 and 16 in a PTBP2-dependent manner (*Licatalosi et al., 2012*; *Li et al., 2014*). Notably, KIS knockdown (shKIS) increased the exclusion of these exons about fourfold (*Figure 3B*). Since the observed changes could be due to changes in the expression of PTBP2 or in one of its interactors, we analyzed the expression levels of PTBP2, MATR3, and hnRNPM. No significant differences were observed between cortical neurons infected with shCtrl vs shKIS (*Figure 3—figure supplement 1C*).

To further confirm the effects of KIS on PTBP2 activity, we used a fluorescent reporter that monitors in vivo AS in cultured neuroblastoma N2A cells. The minigene reporter pFlareG-exon 18 of the *DLG4* gene produces RFP and GFP from the exon-skipping and exon-including isoforms, respectively (*Zheng et al., 2013*). PTBP2 induces the exclusion of exon 18, which control *DLG4* expression during neuronal development (*Zheng et al., 2012*). We found that KIS expression significantly decreased exon 18 exclusion levels compared to non-transfected cells or cells transfected with the K54A kinase dead mutant (*Figure 3C*). These results suggest that KIS inhibits PTBP2 activity on exon 18.

The *DLG4* gene encodes PSD95, a key regulator of synaptic assembly, and the exclusion of exon 18 produces a change in the isoform expression pattern and induces mRNA degradation by NMD (*Zheng et al., 2012*). Thus, we decided to analyze the effects of PTBP2 and KIS on PSD95 endogenous levels in hippocampal neurons. Whereas transfection of wt or a phospho-null triple mutant (3SA) of PTBP2 produced a reduction in PSD95 levels when compared to control neurons, expression of a phosphomimetic mutant (3SD) had no significant effect (*Figure 3D*). In agreement with the above experiments using the splicing fluorescent reporter, KIS overexpression produced an increase in endogenous PSD95 levels compared to control neurons or neurons transfected with the K54A mutant (*Figure 3D*). Together, these results suggest that the phosphorylation state of PTBP2 strongly modulates its role on PSD95 expression. To test the effects of KIS on PTBP2 in this scenario, we co-transfected phosphorylation variants of KIS and PTBP2. First, KIS rescued the negative effects of

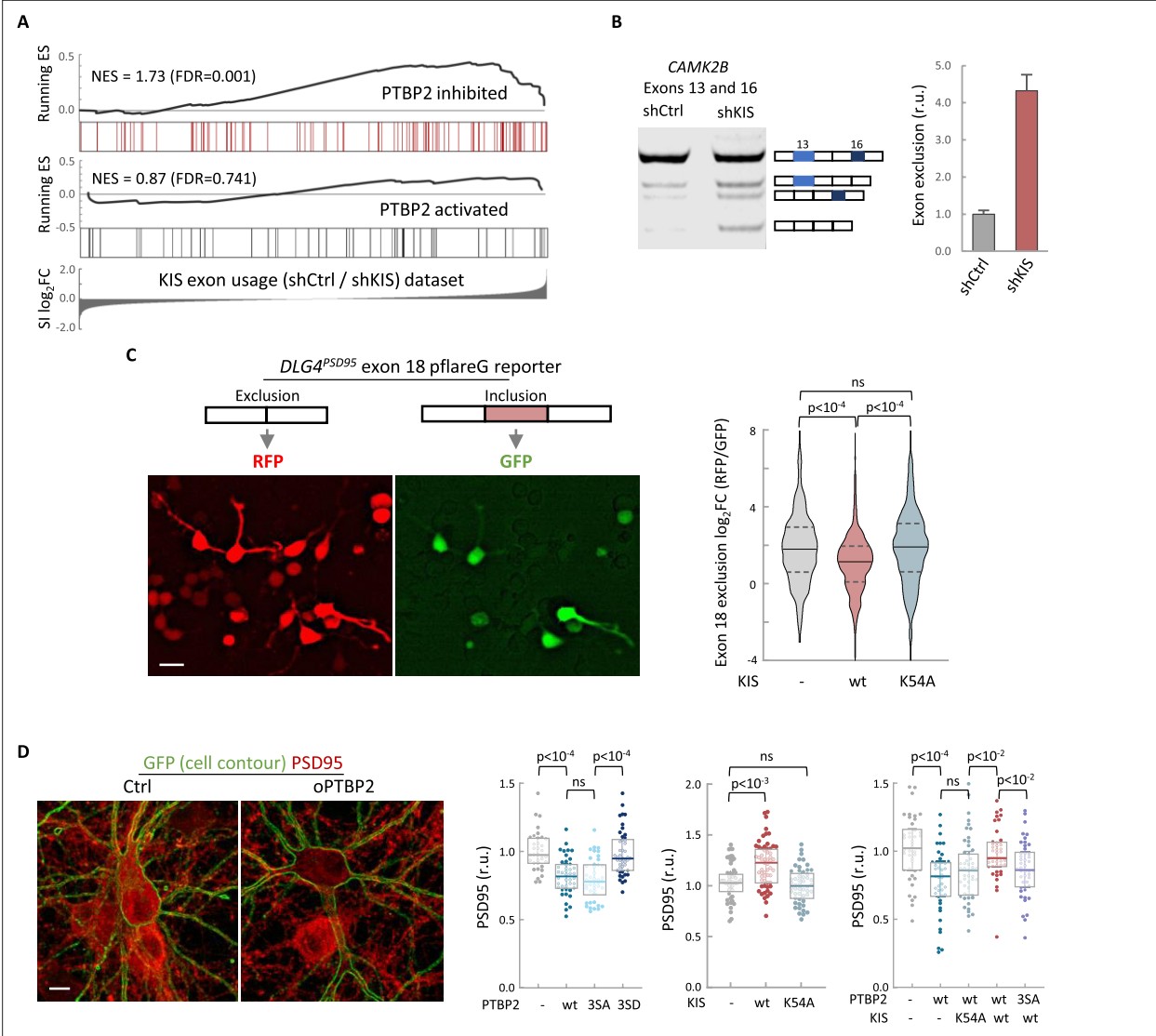

**Figure 3.** KIS kinase activity inhibits Ptbp2-exon-dependent exclusion. (**A**) Gene enrichment analysis of PTBP2-regulated exons. Barcode plots show the position of exons excluded (red) or included (gray) by PTBP2 within the transcriptomic dataset of exons upregulated by KIS. Normalized enrichment scores (NES) and the corresponding false-discovery rate (FDR) values are shown. (**B**) Semiquantitative RT-PCR of *CAMK2B* from control (shCtrl) and KIS knockdown neurons (shKIS). The level of isoforms displaying exon exclusion relative to the full-length isoform is plotted as mean ± standard error of the mean (SEM) ($n = 3$). (**C**) Neuroblastoma N2A cells co-transfected with dual-fluorescent reporter (*Zheng et al., 2013*) and KIS or KIS$^{K54A}$. *DLG4* exon 18 exclusion is plotted as a function of the RFP to GFP fold change (log$_2$FC) from ctrl ($n = 672$), KIS ($n = 1054$), or KIS$^{K54A}$ ($n = 970$) expressing cells. Median ± $Q$ values and the results of Mann–Whitney test are also shown. Scale bar, 20 µm. (**D**) PSD95 immunofluorescence of hippocampal neurons transfected at 5 DIV with vectors expressing PTBP2 and GFP as reporter, and fixed at 12 DIV. Plots show the quantification of PSD95 endogenous levels in single cells transfected with the indicated wild type and mutant proteins in three sets of experiments. Median ± $Q$ values ($n > 40$) and the results of Mann–Whitney tests are also shown. Scale bar, 10 µm.

The online version of this article includes the following source data and figure supplement(s) for figure 3:

**Source data 1.** *CAMK2B* exon exclusion analysis.

**Figure supplement 1.** PTBP2 RNA-seq analysis.

PTBP2 on PSD95 expression levels. Second, this rescue was dependent on the kinase activity of KIS since co-transfection of PTBP2 with the inactive KIS$^{K54A}$ mutant did not increase PSD95 expression. Finally, KIS did not rescue PSD95 levels when co-transfected with the phospho-null 3SA mutant of PTBP2 (*Figure 3D*). Taken together, our findings provide substantial evidence that KIS inhibits PTBP2 activity by phosphorylation.

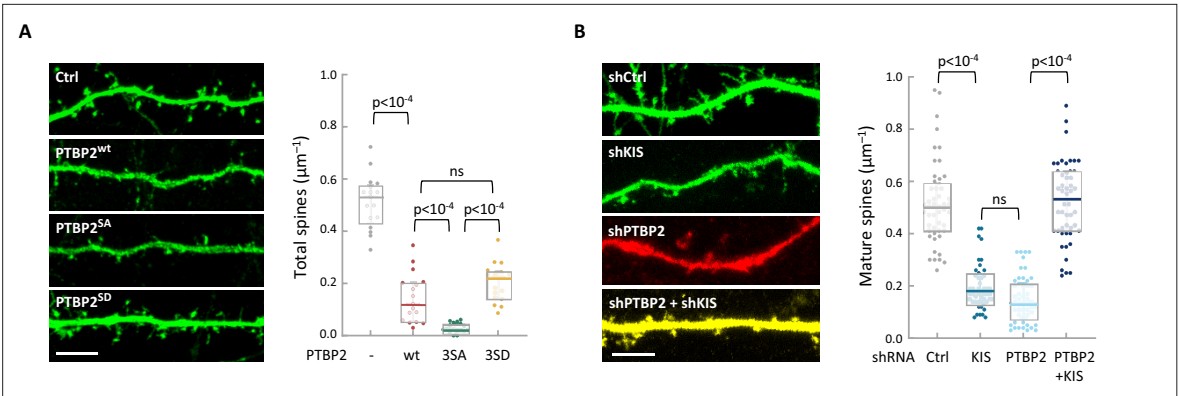

**Figure 4.** The functional interaction between KIS and PTBP2 has an essential role in regulating dendritic spine formation. (**A**) Hippocampal neurons were co-transfected at 5 DIV with a GFP expression plasmid and either a control or PTBP2 expression plasmid. Spine morphology of transfected neurons at 14 DIV was visualized by GFP fluorescence (scale bar, 5 μm). Plot shows the number of spines/μm of 50 neurons per condition from three independent experiments. Median ± $Q$ values and the results of Mann–Whitney tests are also shown. (**B**) Hippocampal neurons were transfected at 5 DIV with lentiviral vectors expressing GFP or RFP and control, KIS, or PTBP2 shRNAs. Spine morphology of transfected neurons at 14 DIV was visualized by GFP fluorescence (scale bar, 5 μm). Plot shows the number of mature spines/μm of 20 neurons/condition from three independent experiments. Median ± $Q$ values and the results of Mann–Whitney tests are also shown.

The online version of this article includes the following figure supplement(s) for figure 4:

**Figure supplement 1.** The interplay between KIS and PTBP2 has an important role in neuronal development.

## KIS and PTBP2 display opposing functional effects on dendritic spine maturation

To study the functional relevance of KIS phosphorylation, we examined the physiological consequences of PTBP2 phosphomutants in differentiated neurons. We transfected dissociated hippocampal cultures with vectors expressing GFP and PTBP2 or PTBP2 phosphomutants and then examined the spine density of transfected neurons via GFP immunofluorescence. Consistent with data reported by other groups (*Zheng et al., 2012*), ectopic expression of PTBP2 caused a marked reduction in spine density compared with control neurons. Overexpression of phospho-null 3SA mutant had a similar effect to wild-type PTBP2, whereas transfection of the phospho-mimetic mutant 3SD resembled control neurons (*Figure 4A*; *Figure 1—figure supplement 1A*). These data support the notion that phosphorylation by KIS counteracts the negative effects of PTBP2 on synaptic spine maturation.

To further confirm the functional relevance of PTBP2 inhibition by KIS we performed knockdown of the two proteins in hippocampal neurons. As seen previously (*Pedraza et al., 2014*), KIS depletion significantly hampered spine maturation. On the other hand, PTBP2 knockdown produced a drastic reduction on the number of mature spines, which was accompanied by a clear deterioration of the neurons transfected with the vector expressing shPTBP2 and DsRed. However, double-knockdown of KIS and PTBP2 attenuated the effects produced by loss of either PTBP2 or KIS (*Figure 4B*; *Figure 4—figure supplement 1B, C*), supporting the mutually counteracting nature of these two proteins in synaptic maturation.

As shown in *Figure 3A*, exons upregulated by KIS were clearly enriched in those inhibited by PTBP2. However, although not significantly enriched, we detected a different group of exons positively regulated by both PTBP2 and KIS. These two groups of exons upregulated by KIS belong to genes associated with calcium ion activity but with different specific functions. Genes with exons downregulated by PTBP2 are more enriched in transmembrane transfer of a calcium ion from intracellular stores whereas genes with exons upregulated by PTBP2 facilitate the diffusion of calcium ions through the postsynaptic membrane. Interestingly, with respect to cytoskeletal terms, the two groups showed clearly different term enrichments. Genes with exons downregulated by PTBP2 were significantly associated with the tubulin cytoskeleton, whereas genes with exons upregulated by PTBP2 were associated with the actin cytoskeleton (*Figure 4—figure supplement 1C*).

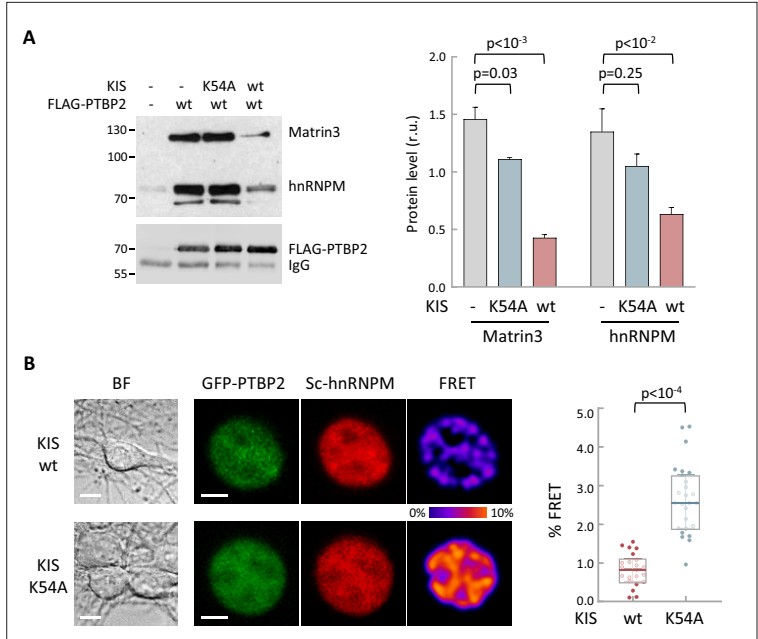

**Figure 5.** Phosphorylation by KIS disrupts PTBP2 protein complexes. (**A**) Constructs expressing FLAG-PTBP2 and KIS, KIS[K54A] were transiently transfected into HEK293T cells, and after 24 hr, cell lysates were subject to FLAG immunoprecipitation. Immunoprecipitates were analyzed by immunoblotting to detect endogenous Matrin3 and hnRNPM proteins. A representative blot is shown with empty vector as control (−). Plots show the quantification of endogenous protein levels relative to FLAG-PTBP2 in immunoprecipitates as mean ± standard error of the mean (SEM) (*n* = 3) values. (**B**) Bright-field (BF), fluorescence and Förster resonance energy transfer (FRET) images of representative nuclei from hippocampal neurons co-expressing GFP-PTBP2 and mScarlet-hnNRPM in combination with KIS or KIS[K54A] proteins (scale bar, 5 µm). BF images are also shown (scale bar, 10 µm). Plot shows FRET levels from single neuron nuclei expressing KIS (*n* = 22) or KIS[K54A] (*n* = 26) as median ± *Q* values. The results of Mann–Whitney test are also shown.

The online version of this article includes the following source data and figure supplement(s) for figure 5:

**Source data 1.** hnRNPM and Matrin3 in FLAG-PTBP2 immunoprecipitates in vivo.

**Figure supplement 1.** In vitro and in vivo experiments showing dissociation of splicing complexes by KIS kinase activity.

**Figure supplement 1—source data 1.** hnRNPM and Matrin3 in FLAG-PTBP2 immunoprecipitates in vitro.

**Figure supplement 1—source data 2.** Representative blot of input samples from *Figure 5A*.

## Phosphorylation by KIS dissociates PTBP2 protein complexes and impairs RNA binding

To further examine how phosphorylation by KIS modifies PTBP2 splicing activity, we decided to investigate whether the inhibition of PTBP2 produced by KIS could be due to alterations of protein–protein interactions within PTBP2 complexes. It is well known that PTBPs proteins are able to bind to different proteins acting synergistically (*Figure 2B*) and, in some cases, modify its specificity for some mRNAs (*Attig et al., 2018*; *Coelho et al., 2016*). On the other hand, in the present study we have shown that key PTBP2 interactors are also KIS phosphotargets. Therefore, it is reasonable to speculate that the association between PTBP2 and its co-regulators could be modulated as a result of phosphorylation by KIS. To assess interactions in vivo, HEK293T cells were co-transfected with FLAG-PTBP2 and HA-KIS or KIS[K54A], and endogenous levels of MATR3 and hnRNPM proteins were analyzed in FLAG immunoprecipitates. Co-transfection with KIS resulted in a significant decrease in PTBP2 binding to MATR3 and hnRNPM. In contrast, this decrease did not occur when the K54A mutant was co-transfected, indicating that the kinase activity of KIS is required to cause dissociation of protein complexes (*Figure 5A*). Similar results were obtained in binding assays in vitro, in which purified KIS forms were added to FLAG-PTBP2 immunoprecipitates, evidencing a direct participation of KIS in the dissociation of PTBP2–MATR3–hnRNPM complexes (*Figure 5—figure supplement 1A*).

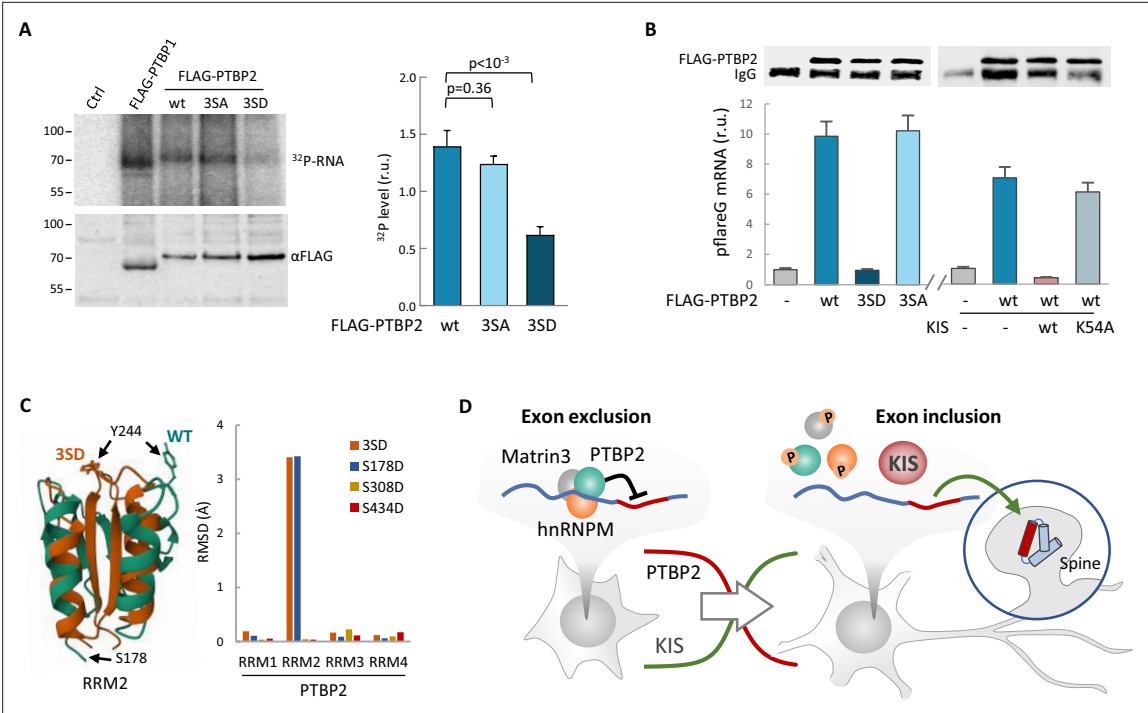

**Figure 6.** Phosphorylation by KIS compromises PTBP2 RNA-binding ability. (**A**) PTBP1 and PTBP2 CLIPs were performed in HEK293T cells expressing FLAG-PTBP1 and FLAG-PTBPs proteins. UV RNA crosslinked and co-precipitated with FLAG-PTBP1 and FLAG-PTBP2s was labeled with 32P-ATP. A representative radiogram gel is shown in the upper image, while the FLAG immunoprecipitates blot is shown in the bottom image. Plot shows the quantification of 32P signal relative to FLAG-PTBP2 immunoprecipitates. Bars represent mean ± standard error of the mean (SEM) (n = 3). (**B**) PTBP2 CLIP was carried out in HEK293T cells transfected with the pFlareG-PSD95 reporter in combination with plasmid vectors expressing the indicated PTBP2 and KIS proteins. UV-crosslinked pFlareG mRNA co-precipitated with FLAG-PTBP2 was analyzed by RT-qPCR and normalized relative to FLAG-PTBP2 in immunoprecipitates (upper blots). Bars represent mean ± SEM (n = 3). (**C**) AlphaFold2 prediction of the RRM2 domain from wild-type (green) and 3SD (orange) PTBP2. The different orientation of Tyr244 is highlighted. Plot shows the root-mean-squared deviation (RMSD) values of the four RRM domains in the different phosphomimetic PTBP2 mutants. (**D**) PTBP2 protein complexes inhibit gene expression through exon exclusion in neuronal precursors whereas, in differentiated neurons, phosphorylation by KIS dissociates PTBP2 protein complexes, which, by being released from the pre-mRNA, allow inclusion of the exon. Since KIS and PTBP2 expression levels are relatively inverted during brain development, phosphorylation-dependent inhibition of PTPB2 by KIS would generate a molecular switch linking transcriptional and alternative exon usage programs during neuronal differentiation.

The online version of this article includes the following source data and figure supplement(s) for figure 6:

**Source data 1.** PTBP2 RNA binding by CLIP.

**Source data 2.** FLAG-PTBP2 immunoprecipitated levels for CLIP analysis in *Figure 6B*.

**Figure supplement 1.** AlphaFold2 predicts structural alterations in the triple 3SD mutants mainly affecting RRM2.

in our experiments co-expressing KIS and PTBP2 in HEK293 cells we did not observe any reduction in splicing factor levels. Next, we wanted to confirm these results using an orthogonal approach, by measuring Förster resonance energy transfer (FRET) between mGFP-PTBP2 and mScarlet-hnRNPM in the nucleus of hippocampal neurons transfected at 6 DIV, when KIS expression has not yet reached maximal levels (*Figure 1B*). In agreement with the data from immunoprecipitation experiments, KIS transfected neurons showed a significant reduction in FRET levels compared to the KIS[K54A] mutant (*Figure 5B*). Similar results were obtained for both hnRNPM and MATR3 when FRET was analyzed in the nuclei of HEK293T transfected cells (*Figure 5—figure supplement 1B*). Thus, the two orthogonal approaches show that phosphorylation by KIS disturbs protein–protein interactions within splicing regulatory complexes.

Finally, we wanted to test whether phosphorylation by KIS could also affect the RNA-binding capacity of PTBP2. For this purpose, we applied the crosslinking and immunoprecipitation cross-linking and immunoprecipitation (CLIP) method to HEK293 cells transfected with FLAG-PTBP2 wt or phosphorylation mutants. Notably, we immunoprecipitated a much lower amount of endogenous RNA crosslinked to the phospho-mimetic mutant 3SD compared to FLAG-PTBP2 wt or 3SA mutant

proteins (*Figure 6A*). Next, we used the same CLIP approach with HEK293 cells transfected with the pFlareG-exon 18 reporter (as in *Figure 3C*), and analyzed levels of the reporter mRNA in PTBP2 immunoprecipitates. In agreement with the previous CLIP experiment, the reporter mRNA levels were strongly reduced in phospho-mimetic 3SD pulldown samples. Moreover, we observed that the reporter mRNA levels in FLAG-PTBP2 immunoprecipitates were reduced when KIS was co-expressed, whereas the KIS$^{K54A}$ mutant had no effect (*Figure 6B*). These results reinforce the notion that phosphorylation of linker regions within PTBP2 may affect the interactions between RRM domains and RNA. Interestingly, AlphaFold2 predicts important structural alterations in the phosphomimetic S178D and triple 3SD mutants particularly affecting RRM2, which has been shown to function as the minimal splicing repressor domain of PTBP2 (*Robinson and Smith, 2006*; *Figure 6C*; *Figure 6—figure supplement 1A*).

## Discussion

Alternative splicing has emerged as a critical and widespread gene regulatory mechanism across organisms and tissues. Among the broad tissue types present in metazoans, the nervous system contains some of the highest levels of AS. AS controls multiple steps of neuronal development and plasticity and is a causal agent of pathology (*Furlanis and Scheiffele, 2018*). We had previously found that KIS is required for the proper expression of postsynaptic proteins at the level of translation (*Cambray et al., 2009*; *Pedraza et al., 2014*) and here we show that KIS plays a key additional role in the regulation of AS, pointing to the existence of concerted mechanisms to fine tune postsynaptic protein expression during neuronal differentiation and synaptogenesis.

Although many documented examples of AS regulation exist, much remains to be understood about the control of specific splicing regulators originating tissue and neuron subtype-specific splicing patterns (*Raj and Blencowe, 2015*). In the present study, we show that phosphorylation by KIS counteracts the activity of the splicing regulator PTBP2 and, as a consequence, controls gene isoform expression during neuronal differentiation and synaptogenesis (*Figure 6D*). Splicing site recognition and pre-mRNA splicing are dynamic processes involving constant rearrangement of ribonuclear proteins on the pre-mRNA being processed. Splicing regulatory proteins contain RNA-binding and protein–protein interaction domains. They bind with low specificity to mostly single-stranded parts of the pre-mRNA. To overcome the low RNA-binding specificity, splicing regulatory proteins use their protein interaction domains to bind to each other. Most splicing factors are post-translationally modified, phosphorylation being an important contributor to the splicing code that governs the fate of a pre-mRNA. Among the best studied classes of splicing proteins regulated by phosphorylation are the SR proteins (serine-/arginine-rich splicing factors) (*Stamm, 2008*). For example, phosphorylation of the SR protein SF2/ASF increases its binding to U1 snRNP and decreases its binding to the RNA export factor TAP/NXF1. Moreover, the ability of its arginine/serine domain (RS) to bind to RNA also depends on its phosphorylation state (*Huang et al., 2004*). The RS domain is highly disordered and switches to a compact, partly ordered arched shape upon phosphorylation (*Thapar, 2015*). Bioinformatics studies have shown that phosphorylation of intrinsically disordered regions occurs more frequently than in folded domains. This could be due to their increased flexibility and therefore improved kinase accessibility. In accordance with this idea, recombinant PTBP1 and PTBP2 under in vitro splicing conditions present different phosphate modifications located in the unstructured N-terminal and linker regions (*Pina et al., 2018*). Although the underlying structural mechanism is not clear, the presence of a phosphoryl group introduces electrostatic charges that can affect the structural stability of binding regions or their ability to interact with RNA or other proteins (*Newcombe et al., 2022*). Supporting this idea, we show that the phosphorylation of PTBP2 linker regions by KIS dissociates both protein–protein and protein–RNA interactions in the PTBP2 complex. Interestingly, AlphaFold2 predicts that a mutation mimicking Ser178 phosphorylation, located at the end of linker 1, would produce major structural changes in the RRM2 domain (*Figure 6C*; *Figure 6—figure supplement 1A*). This domain is of particular relevance for the interaction with RNA and splicing co-regulators, such as Matrin3, Raver1, or hnRNPM (*Coelho et al., 2015*). Within the RRM2 domain, Tyr244 is particularly critical for these interactions (*Joshi et al., 2011*). The AlphaFold2 prediction shows that a phosphomimetic mutation in Ser178 completely changes the spatial orientation of this Tyr244. AlphaFold2 is not yet reliable enough to predict structures with posttranslational modifications, such as glycosylation, methylation, and phosphorylation (*Bertoline*

*et al., 2023*). Thus, we cannot rule out that phosphorylation of other residues (Ser308 and Ser434) causes important structural alterations in PTBP2. In particular, AlphaFold2 fails when the modifications lie within unstructured regions, as is the case of Ser308 (*Chakravarty and Porter, 2022*). The UHM acts as a protein–protein interaction domain (*Kielkopf et al., 2004*) recognizing short peptides called UHM-ligand motifs (ULM). However, among the KIS phosphotargets identified by us, only SUGP1 contained clear ULM candidates (KRKRKSRW385 and KMGW573K). The absence of ULM in PTBP2, Matrin3, and hnRNPM would indicate that phosphorylation is not mediated by stable protein–protein interactions.

It is worth pointing out that Ser178 and Ser434 are also present in PTPB1, suggesting that phosphorylation of PTBP1 by KIS may have a role in tumorigenesis. In gastric cancer, KIS upregulation promotes growth and migration, indicating that KIS functions as an oncoprotein (*Feng et al., 2020*). In hepatocellular cancer, KIS stimulates the expression of specific cell cycle regulation genes. Furthermore, the KIS interactome in liver cancer cells revealed a significant enrichment of potential KIS interactors in mRNA splicing, including hnRNPM and PTBP1 (*Wei et al., 2019*). Finally, KIS has been associated with the epithelial–mesenchyme transition by phosphoproteomic and RNA-seq analyses (*Arfelli et al., 2023*). Thus, in a non-neuronal context, it is plausible that phosphorylation by KIS modulates the capacity of PTBP1 to interact with RNA and co-regulatory proteins to control cell proliferation. One of the new PTBP1 interactors is SUGP2 (SURP and G-patch domain containing 2), which shares high homology with SUGP1. SUGP2 has been shown to interact with the RRM2 region of PTBP1 in HeLa nuclear extracts (*Coelho et al., 2016*; *Coelho et al., 2015*), and SUGP1 interacts and is phosphorylated by KIS in NIH3T3 cells (*Arfelli et al., 2023*). It is worth mentioning that SUGP1 is one of the most enriched proteins in our KIS phosphoproteome (see *Figure 2A*). It forms part of the spliceosome complex, interacts with the general splicing factor U2AF2 and has been reported to play an important role in branch-site recognition by its association with SF3B1. Altered levels of SUGP1 have been found in a large number of uveal melanoma and breast cancer cells (*Zhang et al., 2019*). In the future, it would be interesting to explore whether phosphorylation by KIS interferes with SUGP–PTBP interactions and the functional relevance of this mechanism in cancer and neurogenesis.

The PTBP family represents one of the most salient regulators of splicing in neurogenesis. Sequential changes in the expression of PTBP1 and PTBP2 allow for the establishment of distinct splicing regulatory patterns during neuronal maturation (*Fisher and Feng, 2022*; *Zheng, 2016*). PTBP2 expression decreases during the first postnatal week of brain development, maintaining a moderate level in adulthood (*Fisher and Feng, 2022*; *Li et al., 2007*). Accordingly, our RNA-seq analysis showed significant PTBP2 expression in cortical neurons at 18 DIV (*Figure 2—figure supplement 1A*). Strikingly, PTBP1 knockdown stimulates PTBP2 expression and is sufficient to induce neuronal differentiation of fibroblasts (*Xue et al., 2013*). On the other hand, PTBP2 depletion in developing mouse cortex causes degeneration of this tissue over the first three postnatal weeks, a time when the normal cortex expands and develops mature circuits. Cultured *PTBP2*−/− neurons fail to mature and start to die after 2 weeks in culture (*Li et al., 2014*). However, despite all these data on the physiological relevance of PTBP2 in neurogenesis, the role of this splicing regulator in differentiated neurons and how its activity is regulated were not understood. Regarding neuronal differentiation, it is worth noting that KIS expression increases during brain development (*Bièche et al., 2003*) and in vitro differentiation of hippocampal cultures (*Figure 1B*), coinciding with postnatal decrease of PTBP2 levels (*Zheng et al., 2012*). Therefore, the phosphorylation-dependent inhibition of PTPB2 by KIS and the concerted relative inversion in their levels would generate a molecular switch linking transcriptional and alternative exon usage programs in neuronal development.

In mature neurons, AS has a well-established role in expanding proteome diversity (*Mauger and Scheiffele, 2017*). Although the connection between synaptic activity and the control of KIS expression and/or kinase activity is not yet established, the contraposition of PTBP2 and KIS in splicing may constitute a fine-tuning mechanism to modulate proteome diversity as a function of plasticity-inducing signals. In this regard, single-cell transcriptomic data from hippocampal neurons show that expression variability of KIS and PTBP2 is much higher compared to actin (*Perez et al., 2021*; *Figure 6—figure supplement 1B*). Thus, differences in the expression of these two splicing regulators at a single neuron level would increase protein isoform variability and expand diversity in neural circuits, a crucial property in information processing (*Miller et al., 2019*).

# Materials and methods

**Key resources table**

| Reagent type (species) or resource | Designation | Source or reference | Identifiers | Additional information |
|---|---|---|---|---|
| Strain, strain background (*M. musculus*) | RjOrl:SWISS | Janvier Labs | | Pregnant female |
| Cell line (*H. sapiens*) | HEK 293T | ATCC | CRL-11268 | From human embryo kidney tissue |
| Cell line (*M. musculus*) | Neuro-2a | ATCC | CCL-131 | Mouse neuroblasts isolated from brain tissue |
| Recombinant DNA reagent | pcDNA3-FLAG-PTBP1 | This work | | FLAG tag fused to PTBP1 in pcDNA3 |
| Recombinant DNA reagent | pcDNA3-FLAG-PTBP2 | This work | | FLAG tag fused to PTBP2 in pcDNA3 |
| Recombinant DNA reagent | pEGFP-C3 | Addgene | 6082-1 | N-terminal tagging vector with GFP |
| Recombinant DNA reagent | pEGFP-PTBP2 | This work | | PTBP2 fused to EGFP |
| Recombinant DNA reagent | mScarlet-C1 | Addgene | 85044 | N-terminal tagging vector with Scarlet |
| Recombinant DNA reagent | mScarlet-Matr3 | This work | | Matr3 fused to Scarlet |
| Recombinant DNA reagent | mScarlet-hnRNPM | This work | | hnRNPM fused to Scarlet |
| Recombinant DNA reagent | pGEX-KG | ATCC | 77103 | Bacterial expression vector expressing GST tag |
| Recombinant DNA reagent | pET28a | Novagen | 69864-3 | Bacterial expression vector expressing His tag |
| Recombinant DNA reagent | pcDNA3-FLAG-PTBP1 | This work | | PTBP1 fused to FLAG tag |
| Recombinant DNA reagent | pcDNA3-FLAG-PTBP2 | This work | | PTBP2 fused to FLAG tag |
| Recombinant DNA reagent | pFlareG-PSD95 exon 18 | Addgene | 90266 | Minigene reporter for alternative splicing analysis |
| Recombinant DNA reagent | pCMV-dR8.2 dvpr | Addgene | 8455 | 2nd generation lentiviral packaging plasmid |
| Recombinant DNA reagent | pCMV-VSV-G | Addgene | 8454 | Lentiviral packaging plasmid expressing envelope protein |
| Recombinant DNA reagent | pLKO.3G | Addgene | 14748 | Lentiviral plasmid expressing GFP |
| Recombinant DNA reagent | pLKO.3-DsRed shPTBP2 | This work | | Lentiviral vector expressing DsRed and shRNA sequence against PTBP2 |
| Chemical compound, drug | α-FLAG M2 (monoclonal mouse) affinity gel beads | Merck-Sigma | A2220 | |
| Antibody | α-FLAG M2 (monoclonal mouse) | Merck-Sigma | F3165 | 1:500 (WB) |
| Antibody | α-Matr3 (rabbit polyclonal) | Bionova | A300-591A-T | 1:500 (WB) |
| Antibody | α-hnRNP M3/4 (rabbit polyclonal) | Bionova | A303-910A-T | 1:500 (WB) |
| Antibody | α-mouse IRDye 680RD (goat polyclonal) | LI-COR Biosciences | 926-68070 | 1:1000 (WB) |
| Antibody | α-rabbit HRP (goat polyclonal) | Thermo Fisher | 31460 | 1:1000 (WB) |
| Antibody | α-PSD95 (mouse monoclonal) | Millipore | MABN68 | 1:400 (IF) |
| Antibody | α-SRRM2 (rabbit polyclonal) | Merck-Sigma | HPA066181 | 1:250 (IF) |
| Antibody | α-GFP Alexa 488 (rabbit polyclonal) | Thermo Fisher | A21311 | 1:1000 (IF) |
| Antibody | α-mouse Alexa 568 (donkey polyclonal) | Life Technologies | A10037 | 1:1000 (IF) |
| Antibody | α-rabbit Alexa 568 (rabbit polyclonal) | Life Technologies | A10042 | 1:1000 (IF) |
| Antibody | α-rat Alexa 568 (goat polyclonal) | Invitrogen | A11077 | 1:1000 (IF) |

*Continued on next page*

*Continued*

| Reagent type (species) or resource | Designation | Source or reference | Identifiers | Additional information |
|---|---|---|---|---|
| Commercial assay or kit | CalPhos mammalian transfection kit | Clontech | 631312 | |
| Commercial assay or kit | Lipofectamine 2000 | Invitrogen | 11668030 | |
| Commercial assay or kit | FuGene HD | Promega | E2311 | |
| Commercial assay or kit | NucleoSpin Plasmid Kit | Macherey-Nagel | 740588 | |
| Commercial assay or kit | NucleoBond Xtra Midi Plus EF Kit | Macherey-Nagel | 740422 | |
| Commercial assay or kit | EZNA total RNA purification Kit | Omega | R6834-01 | |
| Software, algorithm | FRETmapJ | This work | | https://www.ibmb.csic.es/en/department-of-cells-and-tissues/control-of-local-mrna-expression/ |
| Software, algorithm | SpineJ | *Pedraza et al., 2014* | | https://www.ibmb.csic.es/en/department-of-cells-and-tissues/control-of-local-mrna-expression/ |
| Software, algorithm | Prism (version 9.5.1) | GraphPad | RRID: SCR_002798 | https://graphpad.com/features |

## Primary cultures

Animal experimental procedures were approved by the ethics committee of the National Research Council of Spain (approval number: B9900083). We have used hippocampal neurons for imaging experiments in which it is important to have isolated mature neurons with well-defined dendritic spines, whereas cortical neuron cultures have been used for biochemical experiments where a larger amount of material is required. Hippocampi and cortex were dissected from E17.5 embryos of Swiss mice (undetermined sex) in Hank's Balanced Salt Solution (HBSS) containing 0.6% glucose and 10 mM 4-(2-hydroxyethyl)-1-piperazineethane sulfonic acid (HEPES). After dissection, tissues were trypsinized (Fisher, 15090046) in HBSS at 37°C for 15 min. Enzymatic digestion was stopped by washing the tissue three times with Minimum Essential Medium Eagle (MEM; Fisher, 31095029) supplemented with 10% fetal bovine serum (FBS) and 0.6% glucose. Hippocampi were left to sediment between washes, and centrifugation was avoided to keep cell viability. Trypsin-treated tissue was then mechanically disaggregated by passing through a flame-polished Pasteur pipette (~10 times). Cells were plated at desired density on poly-D-lysine (Merck-Sigma, P7886)-coated plates (0.5 mg/ml poly-L-lysine in borate buffer, pH 8.5) and maintained in MEM, 10% FBS and 0.6% glucose for 2–4 hr. Culture medium was then substituted for Neurobasal (Fisher, 11570556) supplemented with 2% B27 (Fisher, 11530536) and 1% GlutaMAX (Fisher, 35050-038). Primary hippocampal cultures were transfected using CalPhos mammalian transfection kit (Clontech, 631312) as previously (*Jiang and Chen, 2006*).

## Cell line culture and transfection

HEK293T cells and neuroblastoma N2A cells were cultured in Dulbecco's modified Eagle medium (DMEM) medium with 10% FBS. HEK293T transfection was carried out with Lipofectamine 2000 (Invitrogen, 11668030) and N2A transfection was performed with FuGene HD (Promega, E2311), according to the manufacturer's instructions. Cells were routinely tested for mycoplasma contamination.

## DNA constructs

Site-directed mutagenesis in *PTBP2* and *UHMK1* cDNAs was performed by In-Fusion HD (Takara, 638909). pcDNA3-FLAG, pEGFP-C3, mScarlet-C1, pGEX-KG, pET28a, and pcDNA3-HA were used as host vectors. Plasmids were prepared using NucleoSpin Plasmid Kit (Macherey-Nagel, 740588) for cell line transfections and NucleoBond Xtra Midi Plus EF Kit (Macherey-Nagel, 740422) for neuron transfections.

## Lentivirus production and infection

shRNA sequences in pLKO.3 lentiviral vectors were as follows: GAGTGCGGAGAATGAGTGTTT (TRCN0000027622) against Mm *UHMK1*, and GCTGTACCCTAAGGATTGATT (TRCN0000109231) against Mm *PTBP2*. We used MISSION pLKO.1-puro non-mammalian shRNA control (Merck-Sigma, SHC002). HEK293FT cells were transfected with Lipofectamine 2000 with lentiviral vector, envelope plasmid pVSV-G, and packaging plasmid pCMV-dR8.2 dvpr, and cultured in DMEM 10% FBS. Lentiviruses were harvested 1, 2, and 3 days after transfection, filtered through 0.45 µm cellulose-acetate syringe filters, and concentrated by centrifugation at 26,000 rpm for 2 hr at 4°C. Titration was performed by serial dilution infection in HEK293T and checking the transduction efficiency by fluorescence as described (*Ritter et al., 2017*). Cortical neurons cultured for 7 DIV were infected with 2 Multiplicity of Infection (MOI) lentiviruses for 3 hr, and harvested at 18 DIV.

## RT-PCR and semiquantitative PCR and qPCR

RNAs were extracted from indicated samples using EZNA total RNA purification KIT (Omega, R6834-01), digested with RNase-free DNase (Roche, 11119915001), and 1 µg of RNA was reverse-transcribed into cDNAs using the Prime Script RT Reagent Kit (Takara, RR037B) according to the manufacturer's protocol. RT-PCR primers designed to amplify two spliced isoforms with different sizes are shown in *Supplementary file 3* was performed with Taqman probes (6xFAM-BQ1) on LightCycler@96 Real-Time PCR system (Roche) according to manufacturer's instructions. qPCR primers are shown in *Supplementary file 3*.

## Co-immunoprecipitation, in vitro phosphorylation, and western blot

HEK293T cells were transfected using Lipofectamine 2000 with a proportion 1:3:9 of pEGFP, pcDNA-FLAG/FLAG-PTBP2, and pcNBM470-HA-KIS/HA-KIS$^{K54A}$. After 48 hr post-transfection, cells were washed with cold phosphate-buffered saline (PBS) and homogenized with lysis buffer (50 mM Tris–HCl pH 8, 150 mM KCl, 5 mM MgCl$_2$, 0.25% NP-40, 0.05% sodium deoxycholate, 5% glycerol, 1 mM Ethylene Glycol-bis-(beta-aminoethylether)-N,N,N',N'-tetraacetic Acid [EGTA], 1 mM Ethylenedi-aminetetraacetic Acid [EDTA], and protease and phosphatase inhibitor). To preserve only direct interactions, RNA was digested using RNase I (Fisher, 10330065). After sonication, lysates were clarified by centrifugation at 12,000 × *g* and incubated 2 hr at 4°C with pre-equilibrated α-FLAG M2 affinity gel beads (Merck-Sigma, A2220). Beads were washed three times with lysis buffer. After washes, beads were eluted for western blot analysis or equilibrated to perform in vitro phosphorylation assay. For in vitro assay, FLAG immunoprecipitates were incubated in BK buffer (50 mM HEPES pH 7.6, 10 mM MgCl$_2$, 2 mM dithiothreitol [DTT], 10 µM adenosine triphosphate [ATP]) with 200 ng of His-KIS or His-KIS$^{K54A}$ purified from *E. coli* for 30 min at 30°C. After three washes with lysis buffer, beads were eluded with sodium dodecyl sulfate (SDS) sample buffer. Antibodies used for immunoblotting were: α-FLAG M2 (1:500; Merck-Sigma, F3165), α-Matr3 (1:500; BioNova, A300-591A-T), α-hnRNP M3/4 (1:500; BioNova, A303-910A-T), IRDye 680RD (1:10,000; LI-COR Biosciences, 926-68070), and α-rabbit HRP (1:10,000; Fisher, 31460, RRID: AB_228341).

## In vitro kinase assay

Kinase reactions were carried out in 20 µl of BK buffer (50 mM HEPES pH 7.6, 10 mM MgCl$_2$, 2 mM DTT, 10 µM ATP), 10 µCi of [γ-32P]-ATP, 0.2 µg of GST-KIS (or GST-KIS$^{K54A}$) (*Cambray et al., 2009*), and 2 µg of the corresponding GST-PTBP2 proteins. Kinase reactions were incubated for 20 min at 30°C. Phosphorylated products were separated by SDS–polyacrylamide gel electrophoresis, stained with Coomassie Brilliant Blue, dried, and analyzed by autoradiography.

## PSD95 and SSRM2 immunofluorescence

Hippocampal neurons were fixed at different DIVs using 4% paraformaldehyde and 4% sucrose in PBS for 30 min at 4°C and then washed three times with PBS. Neurons were permeabilized for 5 min with 0.1% Triton X-100 and blocked with 5% Normal Goat Serum (NGS) in PBS (blocking solution). Primary antibody α-PSD95 (EMD Millipore, MABN68, RRID: AB_10807979) were diluted 1:400 in blocking solution. Proteins were detected by incubation with secondary antibodies Alexa 568 donkey anti-mouse (Life Technologies, A10037, RRID: AB_11180865) diluted 1:1000 in blocking solution. Images were acquired with a Zeiss LSM780 confocal microscope using the following parameters:

~15-µm-thick stacks were imaged every 0.5 µm (pinhole set at 1 Airy unit), under ×40 objective (0.13 µm/pixel). Laser power and PMT values were kept constant throughout images and conditions. PSD95 somatic quantification was performed using ImageJ by measuring fluorescent intensity on transfected neurons and relativizing it to the somatic expression of the non-transfected neurons in the same field (*Zheng et al., 2012*). For nuclear SRRM2 immunofluorescence, HEK293T were transfected with GFP or GFP-KIS plasmids. 24 hr post-transfection, cells were fixed with 4% paraformaldehyde for 20 min. Plates were washed twice with PBS and permeabilized with cold 100% methanol for 2 min, before sequential staining with primary and secondary antibodies. Cells were washed three times with PBS and imaged within 24 hr. Primary immunofluorescence was staining with α-SRRM2 (1:250; Merck-Sigma, HPA066181) and secondary with Alexa 568 α-rabbit (1:1000; Life Technologies, A10042; RRID: AB_2534017).

## Spine analysis

Hippocampal neurons were seeded in 4-well plates with 40,000 cells per well, transfected at 4 DIV and fixed at 14 DIV. Primary antibodies α-GFP-Alexa Fluor 488 (Fisher, A21311) and α-Red Rat (Gift from Dr Pons) were diluted 1:1000 in blocking solution. α-Rat Goat Polyclonal Antibody, Alexa Fluor 568 (Invitrogen, A11077) diluted 1:1000 in blocking was used as secondary antibody. Number of fluorescent spines were determined as previously described (*Mendoza et al., 2021*). Images were acquired using a Zeiss LSM780 confocal microscope. Stacks of 10 slices were imaged every 0.37 µm, with a pinhole value of 1 airy unit under a ×63 objective at 0.11 µm per pixel.

## Dual fluorescence labeling

N2A cells were reverse transfected with a proportion 3:9 of pFlareG-PSD95-e18 (*Zheng et al., 2013*) and HA-KIS or HA-KIS$^{K54A}$. After 72 hr fluorescent levels were recorded with a Leica Thunder Imager. Led power and exposition time were kept constant for all images and conditions. After image-deconvolution process, single-cell RFP and GFP were measured with ImageJ. The ratio of exclusion/inclusion of exon 18 in pFlareG was measured as the $\log_2$(RFP/GFP).

## PTBP1 and PTBP2 CLIP assay

HEK293T cells were transfected with FLAG-PTBPs proteins. 48 hr post-transfection, cells were washed once with PBS and crosslinked with 150 mJ/cm$^2$ UV light at 254 nm in Stratalinker 2400. Cells were collected by centrifugation and resuspended on lysis buffer (50 mM Tris–HCl pH 7.4, 100 mM NaCl, 1% IGEPAL CA-630, 0.1% SDS, 0.5% sodium deoxycholate, protease and phosphatase inhibitors). After sonication, lysates where digested with 4 U/ml Turbo DNase (Fisher, AM2238) and 1.5 U/µl RNase I (Fisher, 10330065). Lysates were centrifugated and supernatants incubated with pre-equilibrated α-FLAG M2 affinity gel beads for 2 hr at 4°C. Beads were washed twice with high-salt wash buffer (50 mM Tris–HCl, pH 7.4, 1 M NaCl, 1 mM EDTA, 1% IGEPAL CA-630, 0.1% SDS, 0.5% sodium deoxycholate, protease and phosphatase inhibitors). At that point, beads were equilibrated twice with PNK buffer (20 mM Tris–HCl, pH 7.4, 10 mM MgCl$_2$, 0.2% Tween-20) and RNA 5' ends where labeled with [γ-32P]-ATP and T4 PNK (NEB, M0201L) at 37°C for 5 min. Beads were eluded at 70°C for 5 min with NuPAGE loading buffer (Fisher, 11549166) and loaded in 4–12% NuPAGE Bis-Tris gel (Fisher, 10247002) for subsequent electrophoresis and transference to nitrocellulose membrane (Merck-Sigma, GE10600003). Precipitated RNA was analyzed by autoradiography and FLAG-PTBP protein levels were measured by western blot.

## pFlareG CLIP assay

Protocol used for CLIP assay from pFlareG plasmid was essentially as in the previous section with some modifications. Briefly, HEK293T cells were transfected with pFlareG reporter in combination with plasmid vectors expressing FLAG-PTBP2 proteins and HA-KIS proteins. After crosslinking, cells were resuspended in lysis buffer plus 40 U/ml RNase inhibitor (Attendbio, RNI-G), sonicated and incubated with α-FLAG M2 affinity gel beads overnight at 4°C. Beads were washed twice with lysis buffer, twice with lysis buffer without detergents and once with DNase buffer (10 mM Tris–HCl, pH 7.4, 2.5 mM MgCl$_2$, 0.5 mM CaCl$_2$). Beads were digested with DNase I (Merck-Sigma, DN25) for 15 min and inactivated. RNAs bound to the beads were used as a template for cDNA synthesis using Maxima H Minus cDNA first strand synthesis Kit (Thermo Fisher, 10338179) and random hexamers as primers.

2 μl of cDNA was used as a template in a real-time qPCR assay. pFlareG mRNA was amplified with GFP annealing primers (see **Supplementary file 3**). Relative levels of pFlareG were calculated for each condition using GAPDH as a housekeeping gene and normalized relative to FLAG-PTBP2 protein levels immunoprecipitated checked by western blot using α-FLAG M2 antibody (1:500; Merck-Sigma, F3165) and IRDye 800RD (1:10,000; LI-COR Biosciences, 926-32210).

## FRET imaging

Hippocampal cultures were transfected at 6 DIV with FRET biosensor plasmids expressing mGFP-PTBP2 proteins and pmScarlet-hnRNPM, together with HA-KIS or HA-KIS[K54A], as indicated. Live imaging was conducted 16 hr after transfection. Neurons were live-imaged using a Zeiss LSM780 confocal microscope with a ×40 1.2 NA water-immersion objective. Images were 1024 × 1024 pixels, with a pixel width of 65 nm. Briefly, donor (mGFP-PTBP2) protein was excited at 488 nm, and its emission was measured at 490–532 nm (ID). Excitation of the acceptor (pmScarlet-hnRNPM) was at 561 nm, and emission was measured at 563 to 695 nm (intensity of acceptor [IA]) . We also measured the total signal emitted at 563–695 nm when excited at 488 nm (intensity of FRET [IF]) to obtain the FRET efficiency as $F\% = 100 * (IF - kD*ID - kA*IA)/IA$, kD, and kA, correcting acceptor cross-excitation and donor bleed-through, respectively, with the aid of FRETmapJ, a plugin that also provides maps with the FRET signal as pixel value for local quantification. For HEK293T experiments, cells were transfected with plasmids expressing mGFP-PTBP2 and pmScarlet-hnRNPM or pmScarlet-Matr3, together with HA-KIS or HA-KIS[K54A], as indicated.

## Mass spectrometry-based phosphoproteomics analysis

HEK293T cells were transfected with plasmids expressing KIS and KIS[K54A]. Methods used for sample preparation and data acquisition were as previously described (**Azman et al., 2023**) with some modifications. Briefly, 100 μg of total lysate were used for filter aided sample preparation (iFASP). Digestion with trypsin and chymotrypsin was performed in two different experiments before TMT isobaric labeling (TMT-126TM and TMT-131TM Thermo Fisher). Trypsin (Promega, V5280) and chymotrypsin (Promega, V106A) digested eluates were then pooled independently. The two pooled mixtures were fractionated into four fractions using PierceTM High pH reverse-phase fractionation kit (Life Technologies, 84868), according to the manufacturer's instructions. Fractions samples were subjected to TiO phosphopeptide enrichment using TiO enrichment kit (GL Sciences, 5010-21308), according to the manufacturer's instructions. Phosphopeptides were analyzed using a Q Exactive plus Orbitrap mass spectrometer (Barts Cancer Institute, London). MaxQuant (version 1.6.3.3) software was used for database search and label-free quantification of mass spectrometry raw files. The search was performed against a FASTA file of *Homo sapiens* proteome, extracted from Uniprot (2016). All downstream data analysis was performed using Perseus (version 1.6.2.3). To perform enrichment analysis, using GSEA software (**Mootha et al., 2003**), phospho-site information was summarized at protein level by PPS. PPS was calculated as the sum of the $log_2FC$ value for each phospho-site on a given protein. For GO analysis, the c5.all.v2023.1.Hs.symbols.gmt list provided by GSEA was utilized. Top 50 PTBP2 interactors were obtained from the STRING database by full string network.

## RNA extraction, sequencing, and data analysis

Total RNA was obtained from 18 DIV infected cortical neurons and 7 and 18 DIV of untreated neurons using EZNA total RNA purification kit (Omega, R6834-01), and used to prepare polyA+ enriched libraries for paired-end (2 × 100 bp) sequencing (BGI DNBSEQ platform). More than 90 million reads were obtained per sample, which were aligned to the Mm10 mouse genome using HISAT2 (**Kim et al., 2015**) in the Galaxy platform (**Boekel et al., 2015**). Assignment to transcriptional units, standard filtering steps and basic statistical analysis of triplicate samples was done using Rsubread (version 3.6) (**Liao et al., 2019**) in Bioconductor (**Huber et al., 2015**) and custom R scripts. Genes with a single exon were removed and only exons with more than 5 reads in all samples of one condition were considered. For every annotated exon a splicing index (SI) (**Mauger et al., 2016**) was calculated as the ratio of exon to gene rpkm. GO term and specific GSEA was performed using Enrichr (**Chen et al., 2013**) and GSEA (**Mootha et al., 2003**) tools, respectively. A subset of 8457 exons with low variability (Fano factor <0.0005) within the KIS knockdown triplicate samples was selected, and the significance of differences to control triplicate samples was assessed by calculating the FDR with a

modified *t*-test statistic that incorporates a background variance parameter s0 = 0.1 (*Tusher et al., 2001*). In this manner, a total of 2247 exons offered $q < 0.05$. In contrast, when the same analysis was applied to the experimental dataset randomized by Gaussian attribution or permutation, less than 5 exons displayed $q < 0.05$.

## Statistical analyses

Statistical analyses were performed using GraphPad PRISM (version 9). Pair comparisons were performed with the nonparametric Mann–Whitney test, and the resulting pairwise p values are shown in the corresponding figure panels. Data are displayed as median and quartile (*Q*) values, and the number of samples is described in the figure legends. Protein levels by immunoblotting and mRNA levels by RT-PCR were determined in triplicate samples, and mean ± standard error of the mean values are shown.

## Software

FRETmapJ can be obtained as an ImageJ (Wayne Rasband, NIH) plugin from https://www.ibmb.csic.es/en/department-of-cells-and-tissues/control-of-local-mrna-expression/.

## Acknowledgements

We thank E Rebollo for technical assistance, M Aldea for the helpful comments, and C Rose for editing the manuscript. We also thank members of M Aldea and C Gallego laboratories for the stimulating discussions and ideas. Funding This work was funded by a grant from the Ministry of Science and Innovation of Spain and the European Union (FEDER) (PID2020-113231GB-I00) to CG Medical Research Council UK (MR/P009417/1) and Barts Charity (MGU0346) grants to FKM.

## Additional information

### Funding

| Funder | Grant reference number | Author |
|---|---|---|
| Ministerio de Ciencia e Innovación | PRE2018-083268 | Mónica B Mendoza |
| Ministerio de Ciencia e Innovación | PID2020-113231GB-I00 | Carme Gallego |
| Medical Research Council | MR/P009417/1 | Faraz K Mardakheh |
| Barts Charity | MGU0346 | Faraz K Mardakheh |

The funders had no role in study design, data collection, and interpretation, or the decision to submit the work for publication.

### Author contributions

Marcos Moreno-Aguilera, Conceptualization, Formal analysis, Methodology; Alba M Neher, Mónica B Mendoza, Faraz K Mardakheh, Formal analysis, Methodology; Martin Dodel, Methodology; Raúl Ortiz, Conceptualization, Methodology; Carme Gallego, Conceptualization, Formal analysis, Supervision, Funding acquisition, Investigation, Methodology, Writing - original draft, Writing - review and editing

### Author ORCIDs

Marcos Moreno-Aguilera ![ORCID] https://orcid.org/0000-0001-5243-9582
Faraz K Mardakheh ![ORCID] http://orcid.org/0000-0003-3896-0827
Carme Gallego ![ORCID] http://orcid.org/0000-0001-6961-3524

### Ethics

Animal experimental procedures were approved by the ethics committee of the Research Council of Spain (CSIC).

Decision letter and Author response
Decision letter https://doi.org/10.7554/eLife.96048.sa1
Author response https://doi.org/10.7554/eLife.96048.sa2

## Additional files

### Supplementary files
- Supplementary file 1. RNA-seq datasets from cortical neurons.
- Supplementary file 2. Phosphoproteomic data from HEK293T expressing KIS and KISK54A.
- Supplementary file 3. Primers and vectors used in this study.
- MDAR checklist

### Data availability

The mass spectrometry proteomics data have been deposited to the ProteomeXchange Consortium through the PRIDE partner repository (PXD050320). RNA-seq fastq files and read summaries are available from GEO (GSE260790).

The following datasets were generated:

| Author(s) | Year | Dataset title | Dataset URL | Database and Identifier |
| --- | --- | --- | --- | --- |
| Moreno-Aguilera M, Neher AM, Mendoza MB, Dodel M, Ortiz R, Gallego C | 2024 | KIS counteracts PTBP2 and regulates alternative exon usage in neurons | https://www.ebi.ac.uk/pride/archive/projects/PXD050320 | PRIDE, PXD050320 |
| Moreno-Aguilera M, Neher AM, Mendoza MB, Dodel M, Ortiz R, Gallego C | 2024 | KIS counteracts PTBP2 and regulates alternative exon usage in neurons | https://www.ncbi.nlm.nih.gov/geo/query/acc.cgi?acc=GSE260790 | NCBI Gene Expression Omnibus, GSE260790 |

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
