## [Editor Report]

This well-presented and concise study provides important new insights into the regulation of alternative splicing in neurons by the kinase KIS. The provided data convincingly demonstrates how KIS-mediated phosphorylation of the splicing regulator PTB2 controls exon usage during neuronal differentiation. The findings will be of interest to investigators of splicing regulation, neuronal differentiation and maturation.

---

## [Decision Letter]

[Editors' note: this paper was reviewed by Review Commons.]

---

## [Author Response]

Reviewer #1 (Evidence, reproducibility and clarity (Required)):SummaryIn this manuscript, the authors characterized the molecular function of the brain-enriched kinase KIS by combining transcriptome-wide approaches with molecular and functional studies. They uncover that KIS regulates isoform selection of genes involved in neuronal differentiation and inhibits through phosphorylation the capacity of the splicing regulator PTB2 to interact with both target RNAs and protein partners.Major comments– This is a very clear and well-written manuscript presenting high-quality and carefully controlled experimental results. The authors used an impressive range of approaches (transcriptome-wide exon usage, phospho-proteomic, imaging, biochemical assays..) to profile exon usage alterations upon KIS knock down and provide a mechanistic understanding of how KIS regulate the splicing activity of PTBP2. Specifically, they convincingly demonstrate that the phosphorylation of PTBP2 by KIS leads to both dismantling of PTBP2 protein complexes and impaired RNA binding.My only main concerns relate to the understanding of the biological context in which the mechanism studied may be at play. That KIS can counteract PTB2 activity through direct phosphorylation has been very clearly shown by the authors using overexpression of KIS and /or PTB constructs in different contexts (HEK293T cells, N2A cell line, hippocampal neurons). Whether this occurs endogenously in the context of neuronal differentiation, and how much this contributes to the overall phenotypes induced by KIS inactivation, is less clear. While fully investigating the interplay between KIS and PTB2 in the context of neuronal differentiation is beyond the scope of this study, the three following points could be addressed to provide some evidence in this direction.1. Building on the experiments they perform in a KIS knock-down context (e.g. Figure 3B, or previously described spine phenotype), the authors should investigate whether inhibiting PTBP2 in this context (through shRNA or expression of a phospho-mimetic construct) might suppress the phenotypes observed when inactivating KIS.

As suggested by the reviewer, we have added a new Results section showing the effects on spine maturation in hippocampal neurons expressing PTBP2 phosphomutants and in a PTBP2-KIS double knockdown scenario (Figure 4; Figure 4—figure supplement 1; Results section: P6 L15-P7 L26). First, PTBP2-overexpression effects on post-synaptic protrusion density are exacerbated by the phosphoablated mutant. Intriguingly, the phosphomimetic mutant still has a negative impact in spine formation, suggesting either a residual ability of this protein to interact with its normal partners or the existence of additional roles of PTBP2 in spine development that are Matrin3 and hnRNPM independent. Second, KIS knockdown partially suppresses the defects in mature spine formation produced by the loss of PTBP2. In all, these data support the notion of KIS being a phosphorylation-mediated inhibitor of PTBP2 activity during neuronal differentiation.

2. Based on Figures 1E and 3A, it seems that KIS downregulation affects both exon inclusion and exon skipping, and that its function in exon usage is only partly explained by modulation of PTBP2-dependent exons. Have the authors analyzed the populations of PTBP2-dependent exons that are regulated by KIS in an opposite manner? This may point to specific classes of transcripts (in terms of expression pattern, function, molecular signature) important in the context of endogenous neuronal differentiation.

We have analyzed the GO terms of genes with KIS-upregulated exons by that are either downregulated or upregulated by PTBP2. In both groups we found an enrichment of genes in terms associated with calcium ion activity but with different specific functions. Genes with exons downregulated by PTBP2 are more involved in transmembrane transfer of calcium ions from intracellular stores whereas genes with exons upregulated by PTBP2 facilitate the diffusion of calcium ions through transmembrane postsynaptic. Interestingly, with respect to the cytoskeleton, the two groups show a clearly different term enrichment. Genes with exons downregulated by PTBP2 are significantly associated with the tubulin cytoskeleton, whereas genes with exons upregulated by PTBP2 are associated with the actin cytoskeleton. We have added a paragraph in the Results section (P8 L11-21) and a new panel in Figure 4—figure supplement 1C.

3. The authors should better discuss when and where they think PTBP2 phosphorylation by KIS might be relevant. Is there evidence that this process (or PTBP2 complex assembly) might be regulated upon differentiation or plasticity?

We have modified the Discussion section (P12 L5-L22) as follows:

“Regarding neuronal differentiation, it is worth noting that KIS expression increases during brain development (Bièche et al., 2003) and in vitro differentiation of hippocampal cultures (Figure 1B), coinciding with postnatal decrease of PTBP2 levels (Zheng et al., 2012). Therefore, the phosphorylation-dependent inhibition of PTPB2 by KIS and the concerted relative inversion in their levels would generate a molecular switch linking transcriptional and alternative exon usage programs in neuronal development (Figure 6D).

In mature neurons, alternative splicing has a well-established role in expanding proteome diversity (Mauger & Scheiffele, 2017). Although the connection between synaptic activity and the control of KIS expression and/or kinase activity is not yet established, the contraposition of PTBP2 and KIS in splicing may constitute a fine-tuning mechanism to modulate proteome diversity as a function of plasticity-inducing signals. In this regard, single-cell transcriptomic data from hippocampal neurons show that expression variability of KIS and PTBP2 is much higher compared to actin (Perez et al., 2021) (Figure 6—figure supplement 1B). Thus, differences in the expression of these two splicing regulators at a single neuron level would increase protein isoform variability and expand diversity in neural circuits, a crucial property in information processing (Miller et al., 2019).”

Minor comments1. Figures and associated legends are overall very clear and well-organized. Addressing the following points would however help improving the clarity of some Figures:– In Figure 2EV2C legend, the characteristics of the 3SA constructs are not described

We have modified the legend of Figure 2EV2C (Figure 2—figure supplement 1C Figure in revised version) to clarify this point.

– The difference between Figure EV1A and Figure 1H classifications is unclear, nor the interpretation regarding the different GO classes identified

The gene lists used for the two GO term analyses are different. In Figure EV1A (now Figure 1—figure supplement 1A) the gene list is more restrictive than in Figure 1H as we choose genes with more than one exon upregulated by KIS. In contrast, the analysis in Figure 1H includes all genes with one exon upregulated by KIS.

2. Whether PTBP2 is endogenously the major target of KIS explaining transcriptome-wide changes in exon selection is a possibility that remains to be demonstrated. Thus, the authors should correct and tune down the following sentences:"KIS phosphorylation counteracts PTBP2 activity and thus alters isoform expression patterns …" (end of introduction)"PTBP2 being one of the most relevant phosphotargets" (results, end of the second section)

We agree with the reviewer and we have modified the two sentences (P4 L6-8) and (P6 L13-14) in the revised version of the paper.

Reviewer #1 (Significance (Required)):– The splicing regulator PTBP2 is a known master regulator of neuronal fate whose tightly controlled expression drives the progenitor-to-neuron transition as well as the establishment of neuronal differentiation programs. How this protein is regulated at the post-translational level has so far remains poorly investigated. In this manuscript, the authors provide a thorough mechanistic understanding of how KIS-mediated phosphorylation of PTB2 impacts on its regulation of exon usage. They also provide a transcriptome-wide view on the function of the brain-enriched KIS kinase in exon usage, uncovering its broad functions in alternative splicing.If the physiological context in which KIS-mediated phosphorylation of PTB2 is induced remains to be precisely defined, this work opens interesting new perspectives on regulatory mechanisms at play during neuronal differentiation. Providing extra lines of evidence indicating that KIS acts on neuronal functions through PTBP2 phosphorylation will help further strengthen this aspect.– This manuscript will be of interest to different large communities interested on one hand on the regulation of gene expression programs underlying neuronal differentiation and on the other hand on the molecular regulation of major complexes involved in alternative splicing and isoform selection. It opens new perspectives related to the spatiotemporal regulation of neuronal isoform selection.Reviewer #2 (Evidence, reproducibility and clarity (Required)):SummaryThe manuscript by Moreno-Aguilera et al. shows that the brain enriched protein kinase KIS targets the well known neuronal splicing regulator PTBP2 and several of its interaction partners. As a consequence, PTBP2 activity is down-regulated. Using cultured primary immature neurons they show that KIS expression increases during differentiation and that shRNA knockdown of KIS alters the splicing of many alternative exons. Phosphoproteomic analysis of HEK293 cells transfected with KIS or a kinase dead mutant (K545A) show that it phosphorylates both PTBP2 as well as a cluster of proteins that are known to interact with PTBP2 or its paralog PTBP1. By comparing the new data on KIS-dependent splicing with previous data-sets on PTBP2-dependent splicing targets they show that KIS appears to act antagonistically with PTBP2 when it acts as a repressive regulator, but not when it is an activator. Using combinations of wt and kinase-dead KIS with PTBP2 mutants in the 3 main phosphorylation sites (3SA – non-phosphorylatable, S3D – phosphomimetic) to look at the effects on a known PTBP2 functional target, PSD95, they show that the likely effect of KIS is to antagonise PTBP2 function by phosphorylation at one or more of three residues (S178, S308, S434). Finally, they show that transfected KIS (but not K54A) reduces known protein-protein interactions of PTBP2 and that the triple phosphomimetic PTBP2 mutant shows reduced binding to RNA. Alphafold2 predictions show that the S178 phosphomimetic mutant might alter the conformation of the RRM2 domain, in particular altering the environment of Y244, which has been shown in PTBP1 to be critical for interaction with MATR3 and other coregulators.Major pointsIn general, the conclusions drawn are consistent with the data. I have a few suggestions where the authors could either extend their findings with a few straightforward additional experiments, or clarify some of the existing data.

FigEV4 (also introductory text on p3): RRMs 3 and 4 of PTBP1/2/3 fold as a single back to back packed didomain – with the so-called linker contributing to the didomain fold (e.g. PMID: 24688880, PMID: 16179478) and also extending the RNA binding surface by creating a positive patch (e.g. PMID:20160105 PMID: 24957602). AlphaFold successfully predicts the didomain in full length PTBP2 (https://alphafold.ebi.ac.uk/entry/A0A7I2RVZ4). The authors should therefore use AlphaFold2 to predict the RRM3-4 di-domain structure of wt and phosphomimetic mutant PTBP2s. Phosphorylation of S434 or S434D, which is on the C-terminal end of RRM3 may have no predicted effect on RRM3 alone (FigEV4), but it could conceivably disrupt didomain packing, which could itself have important knock-on consequences for RNA binding. In addition, the inrtoduction of negative charges at S434 might affect the ability of R438, K440 & K441 to interact with RNA. An image of the didomain charge density of WT and mutant PTBP2 would be useful to address this.

As suggested by the reviewer, we have considered the di-domain structure of RRM3 and RRM4, and AlphaFold2 predicted no effects by the phosphomimetic residues. We have added the di-domain predictions to Figure 6—figure supplement 1B.

S434 lies at the very end of RRM3 and limits with a basic region that would not bind RNA in a canonical RRM-dependent manner. In addition, as predicted by AlphaFold2, this basic region is not structured and the effects of a nearby negative charge may be difficult to predict.

Figure 4 could also easily go further in experimentally testing the effects of individual phosphomimetic mutations upon protein-protein interactions (Alphafold predicts that S178D, but not S308 or S434D, should affect Y244 mediated interactions, such as MATR3). The co-IP approach in Figure 4A could readily be used with FLAG-PTBP2 mutants. Likewise, consequences of individual mutations upon RNA binding (Figure 4D) could be tested. The use of a Y244N mutant here would test whether the loss of RNA binding is a consequence of the loss of protein-protein interactions. Such experiments are not essential, but they are readily carried out and have the potential to unravel the consequences of the individual phosphorylation events (more correctly of phosphomimetic mutants).

After building the Y244N mutant we tested PTBP2 interactions with protein partners and observed no significant changes in the levels of Matrin3 and hnRNPM proteins in FLAG-PTBP2 immunoprecipitates nor in the RNA binding ability of PTBP2. These data suggest that, although Y244 is involved in the interaction between PTBP1 and PRI-containing proteins such as Raver1, the interaction between Matrin3 and PTBP2 would involve structural determinants other than the Matrin3 PRI and the PTPB2 Y244 residue. Compared to PTBP1, the nearby flexible loop between RRM2 and RRM3 in PTBP2 is very different and could accommodate specific interaction determinants with Matrin3.

Minor pointsDo KIS regulated exons show enrichment of motifs associated with PTBP2, consistent with the proposed model – particularly CU-rich motifs upstream of exons that are more repressed upon KIS shRNA treatment.

We have not observed a significant enrichment of CU-rich sequences upstream of the top 100 exons upregulated by KIS. Indeed, our data suggest that only a fraction of exons upregulated by KIS are inhibited by PTBP2.

For the splicing analysis pipeline, how were exon-exon junction reads treated? If "only exons with more than 5 reads in all samples" were considered, will this not exclude highly regulated exons that are completely skipped under one condition?

This sentence has been corrected as "only exons with more than 5 reads in all samples of one condition…" (P17 L15)

The Introduction mentions U2AF homology (UHM) domains, but neglects to discuss their known binding partners – ULMs (UHM ligand motifs), which contain an essential tryptophan. It would be useful for the discussion to highlight whether any direct KIS interactors possess ULMs and how this relates to the phospho-targets identified here. The authors may wish to draw the parallel with the structurally analogous way that PTBP1 (and presumably PTBP2) interact with their short peptide ligand motifs.

As suggested by the reviewer, we have searched for ULM sequences in the identified KIS phosphotargets, but we only found clear ULMs in SUGP1, which contains KRKRKSRW^385^ and KMGW^573^K. The absence of ULM motifs in most of the proteins identified in the KIS phosphoproteome would indicate that phosphorylation does not require stable protein-protein interactions. We have added these lines to the Discussion section (P11 L5-L9). We completely agree with the reviewer that, in future work, it would be very interesting to test the possibility that KIS binding modulates the composition and functional properties of splicing complexes through ULM domains.

Figure EV2C. The S3A and S308A mutations clearly reduce phosphorylation. However, the effects of S178A and S434A are far less clear. Presumably the quantitation shown in the lower panel of EV2C relies on normalization to PTBP2 protein input, which appears quite variable in the Coomassie gel. It might be better to repeat the experiment with uniform protein inputs. Minimally, details of the quantitation approach should be added to Materials and methods.

The different levels of reduction in ^32^P incorporation displayed by the single phosphonull mutants suggests that phosphorylation follows a hierarchical pattern, S308-P facilitating phosphorylation of the other two phosphosites. We have added this comment to the revised version of the paper (P5 L27-31). As mentioned by the reviewer, ^32^P incorporation was made relative to the total amount of PTBP2 present in the assay, which was deduced from cold Coomassie-stained gels run in parallel to the radioactive gels with same amount of proteins. We have added details of the quantification in the Methods section from 3 independent experiments

Figure 3D shows PTBP2 overexpression, but the main text (p7) states KIS overexpression.

The text and panel order in Figure 3D were misleading and have been corrected (P7 L14-19).

Figure 4B should have a scale bar for the FRET signal

Done (now Figure 5B)

Figure 4E should indicate the location of S178

Done (now Figure 6C)

Reviewer #2 (Significance (Required)):SignificanceThis interesting, clear and concise manuscript provides important new insights into the way that a neuron specific kinase can regulate neuronal splicing networks by phosphorylating and thereby downregulating the known neuronal splicing regulator PTBP2. Alternative splicing is known to play a particularly important role in neurons, so this demonstration of an additional layer of regulation by post-translational modification should make the manuscript of wide interest to investigators of splicing regulation, neuronal differentiation and maturation.Issues that are not addressed in the manuscript include; (i) how does KIS specifically target PTBP2 and related proteins? The UHM domain can mediate interaction with ULM containing splicing factors (such as U2AF2, SF3B1), but none of the identified targets have known ULMs. (ii) the consequences of individual phoshomimetic mutants upon protein-protein interactions and RNA binding could readily be explored further using computational and experimental methods already used in the manuscript.For context, this reviewer has a direct interest in the mechanisms of regulation of alternative splicing, but not in the context of neurons (though I am familiar with a lot of the relevant literature), and I do not have expertise in neuronal cell biology.Reviewer #3 (Evidence, reproducibility and clarity (Required)):In this manuscript, the authors explored the function of the protein kinase KIS in splicing regulation associated with neuronal differentiation in vitro. KIS is a serine threonine kinase known to phosphorylate splicing factors such as SF1 and SUGP1, and to be preferentially expressed in adult brain in mammals. Using an shRNA based approach, the authors characterize cassette exon usage upon partial KIS depletion in cultured mouse cortical neurons.In parallel using mass spectrometry of proteins in KIS overexpressing HEK293 cells, they identify potential KIS substrates including the splicing regulator PTBP2. They confirm that recombinant KIS can phosphorylates PTBP2 in vitro. They show a correlation between KIS-activated and PTBP2-inhibited exons using published data for this factor. They report opposite effects of KIS and PTBP2 on CamKIIB splicing and Finally, coimmunoprecipitation and FRET experiments suggest that KIS inhibits the interactions of PTBP2 with known protein binders, hnRNPM and Matrin3 as well as with RNA. Altogether these data suggest that KIS downregulates PTBP2 during neuronal differentiation.Major comments:Overall the manuscript is well written and the data are interesting.However several points could have been more extensively studied or discussed to achieve a stronger demonstration of the role of KIS in PTBP2 phosphorylation and neuronal differentiation.1) To minimize possible off target problems, the RNAseq analysis would be more convincing if replicated with a second shRNA to knockdown KIS.

The efficiency of the selected shRNA had been validated both by the supplier (Merck-Σ) and in our previous work, which also included a complementation assay (see Figure 4A-C in Pedraza et al. 2014).

2) Part of the reported splicing changes might reflect an indirect consequence of an altered differentiation contributing to the correlation observed in figure 1F. It would be interesting to confirm splicing changes using shorter incubation times with the shRNA compared to the 11 days used in this study.

The levels of splicing regulators such as PTBP1 and PTBP2 change quite markedly during the initial phases of neuronal differentiation (Zheng et al. 2012). However, we observed no change in their levels when comparing KIS knockdown to control conditions, suggesting no major upstream effects on the differentiation program per se. But, more important, whereas the splicing pattern of CamKIIβ transcript was clearly affected by KIS knockdown at 18 DIV (Figure 3B), we observed no changes at 14 DIV.

3) Standard deviation is more relevant to describe data dispersion in all figures.

For non-parametric statistics we prefer the interquartile range as a measure of dispersion. For the parametric statistics of 3 independent experiments we show the standard error of the mean as a measure of dispersion.

4) Previous papers of the group described a function of KIS in translation (Cambray et al. 2009, Pedraza et al. 2014). This is not discussed here. For example, the possibility that RBPs are regulated by KIS at the translation level is not excluded by the analysis in Figure EV2a.

It is an interesting comment by the reviewer that we have considered during the course of this work. Nevertheless, in our experiments coexpressing KIS and PTBP2 in HEK293 cells we did not observe any reduction in splicing factor levels. We have included a representative immunoblot (Figure 5—figure supplement 1C) of input samples from the experiments shown in the corresponding main figure (Figure 5A in the revised version).

Minor comments:

Figure 1: The authors state that "KIS…accumulates in nuclear sub-structures adjacent to those formed by splicing factors". As the figure presents in fact GFP-KIS, it should be mentioned, and how this localisation is relevant for endogenous KIS should be addressed.

We have corrected the text to mention that GFP-KIS was used to analyse its nuclear localization pattern as shown in Figure 1A (P4 L12-13). We had previously validated the nuclear localization (Boehm et al., 2002) of an N-terminal fusion of KIS to a fluorescent protein (Cambray et al., 2009).

Figure EV1: SI range in panel D is very different from that in panel C and Fig1E.

In this figure we plot the average SI obtained from bins with 2500 exons, which would necessarily narrow the SI range obtained from individual exons. Our data indicate that protein disorder would only constitute a minor, but significant, factor in exon usage.

On page 4 "KIS expression reached maximal levels in hippocampal cultures (Figure 1B)." However the figure legend indicates that this analysis was performed with cortical neurons. The use of cortical or hippocampal neurons along the manuscript should be clarified.

We apologize for the typing mistake, and it has been corrected in the revised version (P4 L17-18)

page 4 " KISK54A, a point mutant without kinase activity" The authors should indicate the reference.

The reference to Maucuer et al. (1997) has been added (P5 L17)

Figure EV2C: It is not clear whether the Coomassie staining and autoradiography do correspond to the same gel.

^32^P incorporation was made relative to the total amount of PTBP2 present in the assay, which was deduced from cold Coomassie-stained gels run in parallel to the radioactive gels with same amount of proteins.

Figure 3C. The authors use a dual fluorescence reporter to analyse PSD95 exon 18 splicing. However the well to well variability in such experiments might be elevated. Not only the cells number in a single well but also the number of replicates should be indicated and well to well variability reported.

As stated in the figure legend, the dual fluorescence reporter experiment has been analyzed at a single-cell level. Using ImageJ software, we analyzed the fluorescence of 1054, 970 and 672 cells expressing KIS, KIS^K54A^ or none, respectively, from 3 independent experiments.

Figure 3D. The precise timing for the transfection and culture of cells before staining is unclear.

Hippocampal neurons were transfected at 5DIV and fixed at 12 DIV. This description has been added to the legend in Figure 3D.

Figure 4A. The input should be loaded to evaluate the coIP efficiencies and ascertain that KIS does not downregulate Matrin3 and hnRNPM levels.

We agree with the reviewer. We have included a representative immunoblot (Figure 5—figure supplement 1C) of input samples from the experiments shown in the corresponding main figure (Figure 5A in the revised version).

Figure EV4A. No difference of Matrin3 binding is to be seen on the gel. In addition, the authors should confirm that PTBP2 or binders are phosphorylated by recombinant KIS. The preparation of GST-KIS is not described.

We agree with the reviewer that in the in vitro assays the differences in phosphorylation are not as clear as in the in vivo experiments. Figure S2C shows an in vitro kinase assay of PTBP2 by recombinant KIS. Finally, we include a reference (Pedraza *et al.,* 2009) for the preparation of recombinant GST-KIS (P14 L14)

Page 6: "We found that PTBP2-inhibited exons are significantly (FDR=0.001) enriched in KIS knockdown neurons, supporting the notion that KIS acts on AS, at least in part, by inhibiting PTBP2 activity." This should be rephrased as in fact PTBP2-inhibited exons are enriched among KIS activated exons.

The sentence has been rephrased as “We found that PTBP2-inhibited exons are significantly (FDR=0.001) among KIS activated exons…” (P6 L20-21)

Page 10: "SUGP1 is one of the most enriched proteins in our KIS phosphoproteome (see Figure 2A)". Phosphorylation and interaction with KIS was already reported by Arfelli and coll. 2023 supplementary figure 2.

We have modified the Discussion section to add this information (P11 L21-22)

Page 10: " It forms part of the spliceosome complex, interacts with the general splicing factor U2AF2 and has been reported to play an important role in branch recognition by its association with SF3B1." A reference is needed there.

A reference to Zhang et al. (2019) has been added.

Page 10: The authors previously reported a differentiation defect in cultured neurons 'Cambray et al., 2008' that was not observed by another group (Manceau et al., PLOS One 2012). This should be discussed in view of these more recent results. Is there any differentiation defect in the experiments reported there?

We have added a new Results section showing the effects on spine maturation in hippocampal neurons expressing PTBP2 phosphomutants and in a PTBP2-KIS double knockdown scenario (Figure 4 and Figure 4—figure supplement 1; Results section: P6 L15-P7 L26). First, PTBP2-overexpression effects on post-synaptic protrusion density are exacerbated by the phosphoablated mutant. Related to the point raised by the reviewer, KIS knockdown also decreased spine emergence and maturation, but partially suppressed the defects produced by the loss of PTBP2. In all, these data support the notion of KIS being a phosphorylation-mediated inhibitor of PTBP2 activity during neuronal differentiation.

Statistical values are difficult to read in the figures. Please use larger fonts.

Done

Reviewer #3 (Significance (Required)):This manuscript brings new elements supporting the function of the protein kinase KIS in splicing regulation in neurons. In particular it identifies for the first time the splicing regulator PTBP2 as a substrate for KIS.It will be of interest to a specialized audience of researchers interested in splicing regulators in neuronal differentiation.